# HaDeMiF: Hallucination Detection and Mitigation in Large Language Models

**Xiaoling Zhou[1], Mingjie Zhang[1], Zhemg Lee[2], Wei Ye[1,\*], & Shikun Zhang[1,\*]**
[1]Peking University [2]Tianjin University

## Abstract

The phenomenon of knowledge hallucinations has raised substantial concerns about the security and reliability of deployed large language models (LLMs). Current methods for detecting hallucinations primarily depend on manually designed individual metrics, such as prediction uncertainty and consistency, and fall short in effectively calibrating model predictions, thus constraining their detection accuracy and applicability in practical applications. In response, we propose an advanced framework, termed HaDeMiF, for detecting and mitigating hallucinations in LLMs. Specifically, hallucinations within the output and semantic spaces of LLMs are comprehensively captured through two compact networks—a novel, interpretable tree model known as the Deep Dynamic Decision Tree (D3T) and a Multilayer Perceptron (MLP)—which take as input a set of prediction characteristics and the hidden states of tokens, respectively. The predictions of LLMs are subsequently calibrated using the outputs from the D3T and MLP networks, aiming to mitigate hallucinations and enhance model calibration. HaDeMiF can be applied during both the inference and fine-tuning phases of LLMs, introducing less than 2% of the parameters relative to the LLMs through the training of two small-scale networks. Extensive experiments conclusively demonstrate the effectiveness of our framework in hallucination detection and model calibration across text generation tasks with responses of varying lengths.

## 1 Introduction

In recent years, large language models (LLMs) have made remarkable advancements, showcasing outstanding performance across a wide range of applications (Schaeffer et al., 2024; Thirunavukarasu et al., 2023; Achiam et al., 2023). Despite their impressive performance, these models remain susceptible to knowledge hallucination (Cohen et al., 2023; Liu et al., 2025; Zhang et al., 2023), that is generating nonfactual responses with unwarranted confidence. This issue undermines user trust and significantly restricts the applicability of LLMs in domains that demand high reliability, such as legal, financial, and educational domains Zhou et al. (2024a). Consequently, detecting and mitigating hallucinations in LLMs has garnered increasing attention from the academic community (Azaria & Mitchell, 2023; Zhang et al., 2024b; Kuhn et al., 2023; He et al., 2025).

Previous studies have proposed various approaches for hallucination detection (Huang et al., 2023; Ji et al., 2023a), targeting either the output space or the internal states of LLMs. For example, predictive confidence and entropy have proven to be effective in detecting hallucinations in natural language processing tasks (Malinin & Gales, 2020; Manakul et al., 2023; Kadavath et al., 2022; Yin et al., 2023; Zhou et al., 2023). Moreover, many studies evaluate hallucinations by leveraging the self-consistency of LLMs across multiple predictions for the same query (Liang et al., 2024; Wang et al., 2023). In contrast to output space-based detection, Chen et al. (2024a) introduced a method that utilizes the internal states of LLMs, capturing divergence and correlation between different sentence representations through the eigenvalues of the covariance matrix. Although these methods have proven to be effective, their reliance on single-aspect indicators, such as uncertainty, consistency, and eigenvalues, confined to either the output space or the internal space, constrains their detection accuracy and hampers their generalizability to more complex scenarios or diverse data distributions (Chen et al., 2024a; Wang et al., 2023; Manakul et al., 2023). Additionally, the

---

*Corresponding to wye@pku.edu.cn; zhangsk@pku.edu.cn.

majority of these methods do not facilitate hallucination mitigation and model calibration, restricting their effectiveness in generation tasks (Manakul et al., 2023; Su et al., 2024; Zhang et al., 2023).

In response to these challenges, this study introduces a comprehensive **Ha**llucination **De**tection and **Mi**tigation **F**ramework called **HADEMIF**, which leverages the rich knowledge embedded in both the output space and internal hidden states of LLMs to identify and address hallucinations. Specifically, two efficient deep networks are employed to detect hallucinations and generate adjustment terms that calibrate the model's probability distribution, thereby achieving both hallucination mitigation and model calibration. First, we propose a novel interpretable tree model, termed the Deep Dynamic Decision Tree (D3T), to detect hallucinations in the output space by leveraging prediction characteristics such as prediction confidence, uncertainty, and consistency, extracted from the LLMs as inputs. As a classification model, D3T predicts whether the generations are hallucinated. It not only benefits from gradient descent training but also provides strong interpretability, enabling the identification of key characteristics that most significantly impact hallucination detection. Additionally, a Multilayer Perceptron (MLP) is employed to capture hallucinations within the deep semantic space, using token hidden states as input. Subsequently, we leverage the outputs from the two hallucination detection networks to calibrate the predictions, aiming to maximize the token probabilities for correct generations while reducing the likelihood of incorrect ones. Our HADEMIF framework can be applied during both the inference and fine-tuning phases of LLMs, introducing less than 2% additional parameters relative to those of the LLMs. To fine-tune LLMs, we further propose an optimization procedure that alternately updates the LLM and the two hallucination detection networks.

The proposed HADEMIF framework is evaluated using the calibration evaluation (CAT) benchmark developed by Liu et al. (2024), in both in-context learning (ICL) Zhou et al. (2024d) and fine-tuning scenarios. This benchmark includes a variety of text generation tasks, with responses differing in length from individual phrases and sentences to full paragraphs. Six popular open-source LLMs are utilized for evaluation: GPT-2 (Radford et al., 2019), GPT-J (Wang & Komatsuzaki, 2021), LLaMA (Touvron et al., 2023a), Llama2 (Touvron et al., 2023b), Llama3 (Dubey et al., 2024), and Vicuna (Chiang et al., 2023), with model sizes ranging from 1.5B to 30B parameters. The experimental results conclusively demonstrate the efficacy of the HADEMIF framework in hallucination detection and model calibration, achieving substantial improvements over existing approaches.

In summary, the primary contributions of our work are as follows:

- We propose an advanced framework, termed HADEMIF, for the detection and mitigation of hallucinations in LLMs. This framework comprehensively captures hallucinations within both the output and internal spaces of LLMs through two compact networks and achieves prediction calibration based on the network outputs.

- We introduce a novel interpretable tree model, named D3T, which is not only trainable via gradient descent but also maintains inherent interpretability. This model provides a clear explanation of the impact of various prediction characteristics, such as uncertainty and consistency, on hallucination detection.

- Our proposed framework can be applied during both the inference and fine-tuning phases of LLMs. For fine-tuning, we outline a detailed optimization process that alternatively updates the LLM and the two hallucination detection networks.

- We conduct extensive experiments on a range of open-source LLMs, covering text generation tasks with varying response lengths. The results consistently demonstrate the effectiveness and broad applicability of our approach in hallucination detection and model calibration, achieving up to a 51% reduction in the average expected calibration error.

## 2 RELATED WORK

**Hallucination Detection** Existing approaches to hallucination detection in the output space typically fall into several categories: performing conventional fact-checking tasks that rely on external knowledge for supervision (Min et al., 2023); assessing model uncertainty, where uncertain outputs are indicative of hallucinations (Xiao & Wang, 2021; Zhou et al., 2022; Yin et al., 2023; Duan et al., 2023); measuring the inconsistency of claims between different LLMs (Cohen et al., 2023; Yang et al., 2023); and evaluating self-consistency (Wang et al., 2023; 2025), where inconsistent outputs often signal hallucinations. Recent research suggests that hallucinations can be traced back to

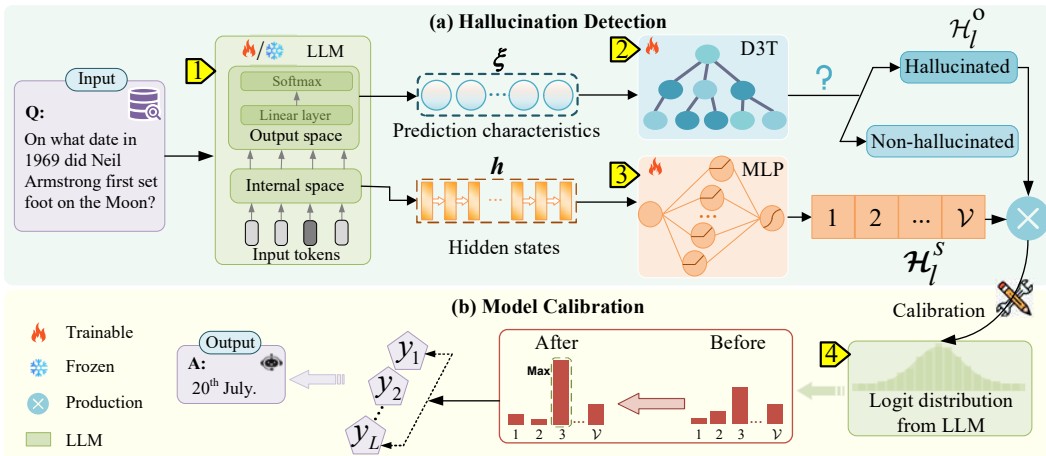

Figure 1: Schematic of the proposed HADEMIF framework. We utilize two efficient networks—a D3T and an MLP—to capture hallucinations within the output and internal spaces of LLMs, respectively. The predictions of LLMs are subsequently calibrated based on the outputs from these two hallucination detection networks, enhancing the reliability of the generated outputs.

learned internal representations (Chen et al., 2024a) and has introduced white-box methods for detecting or predicting hallucinations based on these latent states (Burns et al., 2023; Azadi et al., 2023; Zhu et al., 2024). We argue that both the output and internal spaces of LLMs signal the presence of hallucinations, highlighting the necessity for a comprehensive approach to capture hallucinations across latent states and output transitions throughout the LLM generation process.

**Hallucination Mitigation and Model Calibration**   Hallucination mitigation strategies can be broadly categorized based on the two primary sources: data-related methods and modeling and inference techniques (Ji et al., 2023a; Xin et al., 2024c). Data-related methods aim to refine and augment datasets to ensure the use of more reliable data during training or fine-tuning (Penedo et al., 2023; Zhou et al., 2024b; Chen et al., 2024b). In contrast, modeling and inference techniques are more commonly applied in practical scenarios, as they directly influence the generation process and are not confined to specific tasks or datasets (Touvron et al., 2023a; Ji et al., 2023b; Chuang et al., 2024; Xin et al., 2024a). Within the latter category, model calibration, aiming to align model confidence with the actual probability of output correctness, has proven to be effective for mitigating hallucinations in LLMs (Liu et al., 2024; Zhu et al., 2023), which can generally be divided into post-processing methods (Niculescu-Mizil & Caruana, 2005; Guo et al., 2017) and training-based methods (Pereyra et al., 2017; Xin et al., 2024b; Kapoor et al., 2024). Our method extends this line of research by leveraging hallucination biases captured in the outputs and internal space to calibrate the prediction, thereby mitigating hallucinations and enhancing model calibration.

# 3 METHODOLOGY

This section provides a comprehensive overview of the proposed HADEMIF framework for hallucination detection and model calibration, which is applicable during both the inference and fine-tuning stages of LLMs. Additionally, we describe an online optimization process that involves alternating updates between the LLM and the two hallucination detection networks.

## 3.1 HALLUCINATION DETECTION AND MITIGATION

The HADEMIF framework, as illustrated in Fig. 1, consists of two primary stages: hallucination detection and model calibration. In the first stage, hallucinations are captured within the output and internal spaces of LLMs using two compact neural networks[1] (i.e., D3T and MLP). The second stage addresses hallucinations through logit calibration, guided by the outputs from these two hallucination detection networks.

---

[1]Details of the model complexity analysis are presented in Appendix A.11.

### 3.1.1 HALLUCINATION DETECTION IN THE OUTPUT SPACE

Previous research on hallucination detection has primarily focused on individual aspects such as uncertainty or consistency (Manakul et al., 2023; Liang et al., 2024), which has limited both detection accuracy and broader applicability. Moreover, identifying the useful metrics for effective detection remains a significant challenge. To address these limitations, we propose extracting a comprehensive set of prediction characteristics from the output space of LLMs that effectively capture and reflect hallucinations. These characteristics are subsequently input into a carefully designed deep decision tree, whose inherent interpretability offers valuable insights into the hallucination detection rules associated with prediction characteristics, as well as the relative importance of these characteristics.

**Computational Process of the D3T Model.** D3T is an adaptation of the Deep Neural Decision Tree (DNDT) model (Yang et al., 2018), which is a tree model implemented using a neural network. This method introduces a soft binning function for feature splitting, which is achieved through a linear layer with Softmax as the activation function, and utilizes the Kronecker product operation to determine the final node of the tree. However, DNDT employs a fixed tree structure during training that resem-

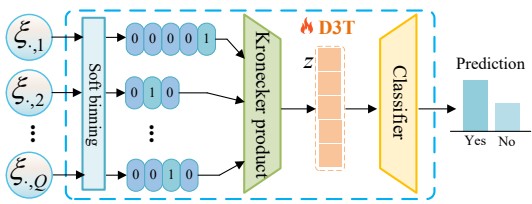

Figure 2: Diagram illustrating the computational process of the D3T model.

bles a perfect $N$-ary tree, where $N$ denotes the number of cut points for each feature. This rigid design often results in numerous redundant nodes, thereby reducing the model's interpretability. In contrast, D3T dynamically learns the optimal number of cut points for each feature throughout training, yielding a more flexible structure that enhances both computational efficiency and interpretability. The calculation process for the D3T model is illustrated in Fig. 2.

Following the approach of DNDT, D3T replaces the traditional hard binning utilized in conventional decision trees with a soft binning function, $\psi(\cdot)$. This function is implemented as a single-layer neural network with a Softmax activation function $\mathbb{S}$:

$$\psi\left(\xi_{\cdot,j}\right) = \mathbb{S}\left[\left(\boldsymbol{w}_j \xi_{\cdot,j} + \boldsymbol{b}_j\right)/\tau\right], \tag{1}$$

where $\xi_{\cdot,j}$ refers to the $j$th prediction characteristic at a time step. The vector $\boldsymbol{w}_j = [1, 2, \cdots, c_j + 1]$ is defined and $c_j$ represents the number of cut points for characteristic $\xi_{\cdot,j}$. In DNDT, the values of $c_j$ are fixed and identical across all characteristics, resulting in a static model structure. Conversely, D3T dynamically optimizes the number of cut points for each feature during training, which is given by $c_j = \lceil C \cdot \sigma(v_j) \rceil$, where $v_j$ is a trainable parameter corresponding to the $j$th feature, $\sigma$ represents the Sigmoid function, and $C$ denotes the constant specifying the maximum number of cut points. During backpropagation, the Straight-Through Estimator (Yin et al., 2019) is employed to circumvent the ceiling operation, a method frequently utilized in the training of activation-quantized neural networks. Furthermore, the trainable vector $\boldsymbol{b}_j$ is defined as $\boldsymbol{b}_j = [0, -\beta_{j,1}, -\beta_{j,1} - \beta_{j,2}, \cdots, -\beta_{j,1} - \beta_{j,2} - \cdots - \beta_{j,c_j}]$, where $\beta_{j,1}$ through $\beta_{j,c_j}$ are the $c_j$ cut points of $\xi_{\cdot,j}$, constrained by the condition $\beta_{j,1} < \beta_{j,2} < \cdots < \beta_{j,c_j}$. The temperature factor $\tau^2$ is also incorporated, and as $\tau \to 0$, $\psi(\xi_{\cdot,j})$ approximates a one-hot vector. For example, if the characteristic $\xi_{\cdot,j}$ is divided into three intervals by the cut points $\beta_{j,1}$ and $\beta_{j,2}$, then the one-hot vector $\psi(\xi_{\cdot,j}) = [0, 1, 0]$ signifies that $\beta_{j,1} < \xi_{\cdot,j} < \beta_{j,2}$.

After binning each characteristic, the Kronecker product is applied to determine the final nodes of the tree:

$$\boldsymbol{z} = \psi\left(\xi_{\cdot,1}\right) \otimes \psi\left(\xi_{\cdot,2}\right) \otimes \cdots \otimes \psi\left(\xi_{\cdot,Q}\right), \tag{2}$$

where $Q$ denotes the number of characteristics. $\boldsymbol{z} \in \mathbb{R}^{\mathrm{d}}$ represents an approximated one-hot vector, indicating the index of the leaf node reached by the extracted characteristics. Subsequently, the vector $\boldsymbol{z}$ is fed into a classifier with weights $\boldsymbol{w}_c \in \mathbb{R}^{\mathrm{d} \times 2}$ to determine whether the current prediction is hallucinated.

**Extraction of Prediction Characteristics.** A series of prediction characteristics $\xi$ are extracted from the output space of LLMs to capture the presence of hallucinations. First, we consider the

---

[2]In applications, we set the value of $\tau$ to 0.1, and the sensitivity analysis regarding this parameter is detailed in Appendix A.8.

commonly used hallucination detection metrics from previous studies, specifically uncertainty and consistency. Next, we incorporate three additional metrics that reflect prediction confidence and are commonly employed in previous machine learning tasks, such as fairness evaluation and sample weighting (Zhang et al., 2020; Ross & Dollár, 2017; Jin et al., 2024; Zhou et al., 2024c). Specifically, all considered metrics are outlined as follows:

- **Probability distribution** reflects the confidence of LLMs in each candidate token. We consider three metrics derived from this distribution: the maximum, minimum, and average values within the probability vector.

- **Uncertainty** is a widely used metric for evaluating token-wise hallucinations, which quantifies the degree of unpredictability in the predictions. It is calculated as $e = \sum_{j=1}^{\mathcal{V}} -p_j \log(p_j)$, where $p_j$ denotes the predicted probability of the $j$th token, and $\mathcal{V}$ represents the vocabulary size.

- **Margin** measures the model's ability to distinguish among different predictions, serving as an indicator of its confidence. We evaluate both the Top-1 and Top-$K$ (set to 10) margins, which are calculated as $\gamma^1 = p^{(1)} - p^{(2)}$ and $\gamma^K = \frac{2}{K(K-1)} \sum_{i=1}^{K-1} \sum_{j=i+1}^{K} (p^{(i)} - p^{(j)})$, respectively, where $p^{(i)}$ denotes the $i$th largest element in the probability vector.

- **Consistency** evaluates the coherence across multiple responses generated by LLMs for the same input. It is quantified using $s = \frac{2}{B(B-1)} \sum_{i=1}^{B} \sum_{j=i+1}^{B} \cos(\boldsymbol{u}^i, \boldsymbol{u}^j)$, where $B$ represents the total number of responses (set to 3), and $\boldsymbol{u}^i$ denotes the logits vector[3] corresponding to the $i$th response.

- **Logits norm** $|\boldsymbol{u}|$ potentially reflects the model's fitting capacity to the input. A larger norm typically signifies that the deep features are more closely aligned with the classifier weights, thereby indicating a higher level of confidence in the prediction.

The aforementioned characteristics are extracted from the LLMs and input into the proposed D3T model to detect hallucinations within the output space.

### 3.1.2 INTERNAL SPACE HALLUCINATION DETECTION AND PREDICTION CALIBRATION

Considering that the internal states of LLMs can also signal hallucinations, the token hidden states[4] are input into an MLP model to capture hallucinations within the deep semantic space. Unlike previous methods that manually define functions to associate internal states with hallucinations (Chen et al., 2024a; Zhu et al., 2024), this approach leverages the universal approximation capability of deep neural networks (Lu & Lu, 2020) to automatically learn the mapping between hidden states and hallucinations. The output of the MLP network, denoted as $\mathcal{H}_l^s(\boldsymbol{h}; \boldsymbol{\Omega}s) \in \mathbb{R}^{\mathcal{V}}$, is then utilized for prediction calibration, where $\boldsymbol{h}$ represents the hidden states of tokens and $\boldsymbol{\Omega}_s$ denotes the MLP parameters. Specifically, the logits of LLMs are calibrated using the outputs from the two hallucination detection networks, as outlined below:

$$\hat{p}\left(y_l \mid \boldsymbol{x}, y_{<l}, \mathcal{H}_l^s, \mathcal{H}_l^o\right) = \frac{\exp(\boldsymbol{u}_l^{(y_l)} - \mathcal{H}_l^o \log(\mathcal{H}_{l,y_l}^s))}{\sum_{v=1}^{\mathcal{V}} \exp(\boldsymbol{u}_l^{(v)} - \mathcal{H}_l^o \log(\mathcal{H}_{l,v}^s))}, \tag{3}$$

where $\boldsymbol{u}_l$ denotes the logits vector at step $l$. Furthermore, $\mathcal{H}_l^o$ represents the probability that the D3T model classifies a prediction as a hallucination, which controls the intensity of the calibration.

Notably, our method does not require any additional hallucination annotations. It only requires training the two hallucination detection networks by minimizing the loss of LLMs on the original training set, with predictions computed as outlined in Eq. (3). This process aims to maximize the token probabilities for correct generations while reducing the likelihood of incorrect ones. Subsequently, the trained D3T and MLP models can be directly employed for hallucination detection and prediction calibration during the inference phase of LLMs.

---

[3]Since logits are the unnormalized scores produced by the final linear layer and directly influence the model's outputs, we regard them as a characteristic of the output space.

[4]We utilize the hidden states preceding the logits, and a comparative analysis of the internal states across different layers is provided in Appendix A.8.

## 3.2 Optimization Procedure for Fine-Tuning LLMs

To incorporate our framework into the fine-tuning phase of LLMs, we propose an optimization process that alternately updates both the LLM and the two hallucination detection networks. Let the LLM be parameterized by $\boldsymbol{\Theta}$. The optimization process proceeds as follows: first, the LLM parameters $\boldsymbol{\Theta}$ are updated using stochastic gradient descent on a mini-batch of training samples $(\boldsymbol{x}_i, \boldsymbol{y}_i)_{i=1}^{n}$, according to the following objective function:

$$\boldsymbol{\Theta}^{(t+1)} \leftarrow \boldsymbol{\Theta}^{(t)} - \eta_1 \frac{1}{n} \sum_{i=1}^{n} \nabla_{\boldsymbol{\Theta}} \left\{ -\sum_{l} \log \hat{p} \left( y_{i,l} \mid \boldsymbol{x}_i, y_{i,<l}, \boldsymbol{\mathcal{H}}_{i,l}^{s}(\boldsymbol{\Omega}_{s}^{(t)}), \mathcal{H}_{i,l}^{o}(\boldsymbol{\Omega}_{o}^{(t)}); \boldsymbol{\Theta}^{(t)}) \right) \right\},$$
(4)

where $\eta_1$ denotes the step size used for updating the LLM parameters.

Subsequently, utilizing the optimized $\boldsymbol{\Theta}^{(t+1)}$, the parameters of the two hallucination detection networks—denoted as $\boldsymbol{\Omega}o$ for D3T and $\boldsymbol{\Omega}s$ for MLP—can be updated as follows:

$$\boldsymbol{\Omega}_{s}^{(t+1)} \leftarrow \boldsymbol{\Omega}_{s}^{(t)} - \eta_2 \frac{1}{n} \sum_{i=1}^{n} \nabla_{\boldsymbol{\Omega}_{s}} \left\{ -\sum_{l} \log \hat{p} \left( y_{i,l} \mid \boldsymbol{x}_i, y_{i,<l}, \boldsymbol{\mathcal{H}}_{i,l}^{s}(\boldsymbol{\Omega}_{s}^{(t)}), \mathcal{H}_{i,l}^{o}(\boldsymbol{\Omega}_{o}^{(t)}); \boldsymbol{\Theta}^{(t+1)}) \right) \right\},$$
(5)

$$\boldsymbol{\Omega}_{o}^{(t+1)} \leftarrow \boldsymbol{\Omega}_{o}^{(t)} - \eta_2 \frac{1}{n} \sum_{i=1}^{n} \nabla_{\boldsymbol{\Omega}_{o}} \left\{ -\sum_{l} \log \hat{p} \left( y_{i,l} \mid \boldsymbol{x}_i, y_{i,<l}, \boldsymbol{\mathcal{H}}_{i,l}^{s}(\boldsymbol{\Omega}_{s}^{(t)}), \mathcal{H}_{i,l}^{o}(\boldsymbol{\Omega}_{o}^{(t)}); \boldsymbol{\Theta}^{(t+1)}) \right) \right\},$$
(6)

where $\eta_2$ represents the step size for updating the parameters of the two hallucination detection networks. To facilitate efficient fine-tuning of the LLMs, we utilize LoRA (Hu et al., 2022), which enables the fine-tuning process to be conducted on a single GPU.

## 4 Experimental Investigation

### 4.1 Datasets

Following Liu et al. (2024), our work evaluates text generation tasks with responses of varying lengths. Specifically, we utilize the CAT benchmark (Liu et al., 2024), which encompasses tasks with responses at the phrase, sentence, and paragraph levels. The phrase-level generation datasets include NaturalQuestions (**NQ**), **SciQ**, and **TriviaQA**, each of which features short responses, such as named entities. For sentence-level responses, we consider **TruthfulQA** and **WikiQA**, where the model outputs full sentences. For paragraph-level tasks, we incorporate **BioGen** and **WikiGen** (Liu et al., 2024). In the BioGen task, LLMs are prompted to write biographies of various figures (Min et al., 2023), with ground-truth answers extracted from corresponding Wikipedia passages. In the WikiGen task, LLMs generate Wikipedia-style descriptions of entities, based on the fact verification dataset FEVER (Thorne et al., 2018). Comprehensive statistics, detailed descriptions of the training and test set construction, as well as illustrative examples for all datasets are presented in Appendix A.1.

### 4.2 Evaluation Metrics

To ensure a fair comparison, the methodology for evaluating the model's confidence in its generated outputs, along with the accuracy of these outputs, follows the approach established by Liu et al. (2024). Specifically, for phrase- and sentence-level tasks, the model's confidence $p_{\boldsymbol{y}}(\boldsymbol{x})$ is calculated as the geometric mean of the sequence of token probabilities:

$$p_{\boldsymbol{y}}(\boldsymbol{x}) = \sqrt[L]{\prod_{l=1}^{L} p\left(y_l \mid \boldsymbol{x}, \boldsymbol{y}_{<l}\right)}.$$
(7)

Additionally, GPT-4 Achiam et al. (2023) is employed to evaluate the correctness of model outputs by determining the semantic equivalence between the generated text and the reference. For paragraph-level tasks, the assessment of accuracy and confidence involves four steps: claim extraction, span mapping, confidence estimation, and correctness estimation (Liu et al., 2024). We then utilize three metrics to evaluate the effectiveness of our approach in hallucination detection and model calibration.

| Task | Metric | | Original LLM | Model Calibration | | | | | HADEMIF | HADEMIF w/ Fine-Tuning |
| | | | | Label Smoothing | Temp. Scaling | LITCAB | LITCAB w/ Temp. Scaling | Calibration-Tuning | | |
|---|---|---|---|---|---|---|---|---|---|---|
| *Phrase Level* | | | | | | | | | | |
| NQ | acc@50 | ↑ | 0.288 | 0.208 | 0.288 | 0.300 | 0.300 | 0.310 | 0.315 | **0.355** |
| | cov@50 | ↑ | 0.115 | 0.061 | 0.115 | 0.105 | 0.105 | 0.115 | 0.115 | **0.120** |
| | ECE | ↓ | 0.171 | 0.186 | 0.165 | 0.101 | 0.083 | 0.051 | 0.034 | **0.026** |
| | Brier | ↓ | 0.196 | 0.212 | 0.193 | 0.169 | 0.164 | 0.142 | **0.116** | 0.119 |
| SciQ | acc@50 | ↑ | 0.764 | 0.212 | 0.764 | 0.762 | 0.762 | 0.761 | 0.760 | **0.766** |
| | cov@90 | ↑ | 0.211 | 0.003 | 0.211 | 0.221 | 0.221 | 0.224 | **0.230** | 0.228 |
| | ECE | ↓ | 0.094 | 0.391 | 0.091 | 0.084 | 0.082 | 0.081 | 0.083 | **0.076** |
| | Brier | ↓ | 0.203 | 0.386 | 0.202 | 0.203 | 0.203 | 0.202 | 0.201 | **0.200** |
| TriviaQA | acc@50 | ↑ | 0.500 | 0.302 | 0.500 | 0.478 | 0.478 | 0.482 | 0.480 | **0.501** |
| | cov@60 | ↑ | 0.111 | 0.019 | 0.111 | 0.201 | 0.201 | 0.222 | 0.234 | **0.240** |
| | ECE | ↓ | 0.112 | 0.184 | 0.079 | 0.081 | 0.079 | 0.080 | 0.080 | **0.075** |
| | Brier | ↓ | 0.203 | 0.259 | 0.195 | 0.203 | 0.199 | 0.193 | 0.190 | **0.185** |
| *Sentence Level* | | | | | | | | | | |
| TruthfulQA | acc@50 | ↑ | 0.314 | 0.181 | 0.314 | 0.314 | 0.314 | 0.386 | 0.415 | **0.430** |
| | cov@40 | ↑ | 0.136 | 0.000 | 0.136 | 0.195 | 0.195 | 0.393 | 0.500 | **0.510** |
| | ECE | ↓ | 0.138 | 0.134 | 0.161 | 0.105 | 0.103 | 0.095 | 0.087 | **0.058** |
| | Brier | ↓ | 0.218 | **0.175** | 0.240 | 0.206 | 0.203 | 0.198 | 0.193 | 0.176 |
| WikiQA | acc@50 | ↑ | 0.388 | 0.273 | 0.388 | 0.397 | 0.397 | 0.441 | 0.629 | **0.653** |
| | cov@50 | ↑ | 0.012 | 0.000 | 0.012 | 0.062 | 0.062 | 0.162 | 0.330 | **0.338** |
| | ECE | ↓ | 0.075 | 0.155 | 0.066 | 0.075 | 0.074 | 0.070 | 0.068 | **0.055** |
| | Brier | ↓ | 0.212 | 0.239 | 0.222 | 0.212 | 0.210 | 0.210 | 0.211 | **0.208** |
| **Average** | acc@50 | ↑ | 0.451 | 0.235 | 0.451 | 0.450 | 0.450 | 0.476 | 0.520 | **0.541** |
| | ECE | ↓ | 0.118 | 0.210 | 0.112 | 0.089 | 0.084 | 0.075 | 0.070 | **0.058** |
| | Brier | ↓ | 0.206 | 0.254 | 0.210 | 0.199 | 0.196 | 0.189 | 0.182 | **0.178** |

Table 1: Comparison between HADEMIF and model calibration methods on the CAT benchmark for phrase- and sentence-level responses. For each metric and dataset, the top scores are highlighted in **bold**, and the second-best scores are underlined. Scores where HADEMIF surpasses both the original LLM and all baselines are highlighted in blue, while those outperforming only the original LLM are marked in green. The final rows summarize the average values across all tasks. Our proposed HADEMIF approach consistently enhances model performance.

- In line with prior research (Guo et al., 2017; Tian et al., 2023; Liu et al., 2024), we employ the **Expected Calibration Error** (ECE) to measure the discrepancy between a model's confidence and its actual accuracy. Specifically, model predictions are grouped according to confidence levels, and we compute the accuracy $acc(b_i)$ and the average confidence $conf(b_i)$ within each bin $b_i$. The ECE is then calculated as $ECE = \sum_i \frac{|b_i|}{M}|acc(b_i) - conf(b_i)|$, where $M$ represents the total number of model outputs. A lower ECE signifies better calibration, indicating a closer alignment between the model's confidence and its actual accuracy.

- The **Brier Score** (Brier, 1950) is a metric commonly used to evaluate tasks that require assigning probabilities to a set of mutually exclusive discrete outcomes or classes, which can be either binary or categorical. Following Liu et al. (2024), we compute the Brier Score as the mean squared difference between the model confidence $p_{\boldsymbol{y}}$ and the binary correctness $I(\boldsymbol{y})$ of its predictions $Brier = \frac{1}{M} \sum_{\boldsymbol{y}}[p_{\boldsymbol{y}} - I(\boldsymbol{y})]^2$. This metric offers a direct assessment of the quality of model calibration.

- Given the importance of model confidence, we also assess model performance using two **selective classification metrics** as detailed in (Liu et al., 2024). The first metric, accuracy at coverage (acc@q), evaluates the precision of the model by examining the accuracy of the top-$q$ percent of predictions. The second metric, coverage at accuracy (cov@p), measures recall by identifying the largest proportion of the most confident predictions where accuracy surpasses a designated threshold $p$. Unlike AUROC (Bradley, 1997), which primarily assesses the quality of confidence scores, these metrics provide a direct evaluation of the model's capability to filter out incorrect predictions by applying specific thresholds.

## 4.3 COMPARED BASELINES

We compare the HADEMIF framework with four traditional and advanced model calibration methods. **Temperature Scaling** (Liang et al., 2018) adjusts the logits by a temperature parameter before applying the Softmax function. **Label Smoothing** (Szegedy et al., 2016) involves fine-tuning the LLMs using LoRA with label smoothing techniques applied during training. **Lightweight Calibration** (LITCAB) (Liu et al., 2024) employs a single linear layer to process the input text representa-

| Task | Metric | | Original LLM | Model Calibration | | | | HADEMIF | HADEMIF w/ Fine-Tuning |
|------|--------|---|--------------|-------------------|---|---|---|---------|------------------------|
| | | | | Label Smoothing | Temp. Scaling | LITCAB | LITCAB w/ Temp. Scaling | | |
| BioGen | acc@50 | ↑ | 0.347 | 0.334 | 0.347 | 0.354 | 0.354 | **0.362** | 0.358 |
| | cov@40 | ↑ | 0.066 | 0.059 | 0.066 | 0.148 | 0.148 | 0.159 | **0.160** |
| | ECE | ↓ | 0.169 | 0.196 | 0.246 | 0.166 | 0.243 | 0.164 | **0.160** |
| | Brier | ↓ | 0.269 | 0.284 | 0.313 | 0.267 | 0.308 | 0.268 | **0.255** |
| WikiGen | acc@50 | ↑ | 0.876 | 0.860 | 0.876 | 0.872 | 0.872 | 0.875 | **0.884** |
| | cov@80 | ↑ | 0.745 | 0.733 | 0.745 | 0.756 | 0.756 | 0.760 | **0.774** |
| | ECE | ↓ | 0.045 | 0.075 | 0.049 | 0.037 | 0.065 | 0.040 | **0.032** |
| | Brier | ↓ | 0.172 | 0.187 | 0.173 | 0.171 | 0.174 | 0.167 | **0.161** |
| **Average** | acc@50 | ↑ | 0.612 | 0.597 | 0.612 | 0.613 | 0.613 | 0.619 | **0.621** |
| | ECE | ↓ | 0.107 | 0.136 | 0.148 | 0.102 | 0.154 | 0.102 | **0.096** |
| | Brier | ↓ | 0.221 | 0.236 | 0.243 | 0.219 | 0.241 | 0.218 | **0.208** |

Table 2: Comparison between HADEMIF and model calibration methods on the CAT benchmark for paragraph-level responses. Our approach consistently enhances the model performance of LLMs in paragraph generation tasks.

tion and predict a bias term, which is then added to the output logits of the LLMs. **Calibration-Tuning** (Kapoor et al., 2024) fine-tunes LLMs by designing a task that enables the model to autonomously evaluate whether its generated responses are consistent with the true answers.

Additionally, we compare HADEMIF with recent methods specifically designed for hallucination detection and mitigation in LLMs. **Verbalization** involves prompting the LLM to self-report its confidence level for a given output, using the prompt provided by Tian et al. (2023). **P(IK)** (Kadavath et al., 2022) introduces a linear layer on top of the LLM's final hidden state corresponding to the last token of a question, training this layer to predict the model's likelihood of accurately answering the question. **Self-Consistency** (Tian et al., 2023; Xiong et al., 2024) operates on the principle that confident responses are more likely to recur when sampling from the model. **Refusal-Aware Instruction Tuning** (R-Tuning) (Zhang et al., 2024a) fine-tunes LLMs on refusal-aware datasets to equip the models with the capability to generate refusal-aware responses, thereby decreasing hallucinations. It is important to note that the three hallucination detection methods and Calibration-Tuning produce only a single aggregated score for the entire generated output, which prevents them from generating scores at the individual claim level. This limitation renders them unsuitable for paragraph-level tasks, where the generated content typically comprises multiple claims. Consequently, following Liu et al. (2024), we exclude these methods from paragraph-level tasks.

## 4.4 EXPERIMENTAL SETTINGS

Our experimental setups follow those outlined in Liu et al. (2024). Specifically, we select Llama2-7B (Touvron et al., 2023b) as the primary backbone model, given its strong performance across a wide range of benchmark datasets. Additionally, we include seven other popular LLMs, ranging in size from 1.5B to 30B. Due to space limitations, we present the results for Llama2-7B in the main text, with results for the other models available in the Appendix. The training process begins with an initial learning rate of $1 \times 10^{-3}$ for both the MLP and D3T networks, which is reduced by a factor of 0.1 at the 20th and 40th epochs. Training is conducted in 50 epochs with early stopping. For fine-tuning the LLMs, the two hallucination detection networks are first trained for 40 epochs, after which an alternating optimization process is applied between the LLMs and the two detection networks. The LLMs are fine-tuned for 5 epochs using LoRA[5] with a rank of 8 and a learning rate of $3 \times 10^{-4}$. More detailed experimental settings are presented in the Appendix.

## 4.5 MAIN RESULTS

**Comparison with Model Calibration Methods** The results of HADEMIF, along with those of four traditional and advanced model calibration methods, are presented in Tables 1 and 2, where some results are from the LITCAB (Liu et al., 2024) paper. **HADEMIF consistently enhances the performance of the original LLM across all tasks, demonstrating its effectiveness in model calibration**. Specifically, HADEMIF reduces the ECE and Brier scores by 51% (from 0.118 to 0.058) and 14% (from 0.206 to 0.178), respectively, for phrase- and sentence-level tasks compared to the

---

[5] https://github.com/microsoft/LoRA

| Task | Metric | | Original LLM | Hallucination Detection and Mitigation | | | | HADEMIF | HADEMIF w/ Fine-Tuning |
|------|--------|---|--------------|-------|---------------|-------------------|----------|---------|------------------------|
| | | | | P(IK) | Verbalization | Self-Consistency | R-Tuning | | |
| *Phrase Level* | | | | | | | | | |
| NQ | acc@50 | ↑ | 0.288 | 0.286 | 0.254 | 0.340 | 0.293 | 0.315 | **0.355** |
| | cov@50 | ↑ | 0.115 | 0.000 | 0.055 | **0.217** | 0.084 | 0.115 | 0.120 |
| | ECE | ↓ | 0.171 | 0.158 | 0.516 | 0.145 | 0.156 | 0.034 | 0.026 |
| | Brier | ↓ | 0.196 | 0.204 | 0.468 | 0.163 | 0.201 | **0.116** | 0.119 |
| SciQ | acc@50 | ↑ | 0.764 | 0.656 | 0.660 | 0.744 | 0.692 | 0.760 | **0.766** |
| | cov@90 | ↑ | 0.211 | 0.004 | 0.117 | 0.124 | 0.119 | **0.230** | 0.228 |
| | ECE | ↓ | 0.094 | 0.188 | 0.318 | 0.101 | 0.190 | 0.083 | 0.076 |
| | Brier | ↓ | 0.203 | 0.276 | 0.344 | 0.227 | 0.285 | 0.201 | **0.200** |
| TriviaQA | acc@50 | ↑ | 0.500 | 0.372 | 0.404 | 0.446 | 0.400 | 0.480 | **0.501** |
| | cov@60 | ↑ | 0.111 | 0.023 | 0.053 | 0.079 | 0.063 | 0.234 | **0.240** |
| | ECE | ↓ | 0.112 | 0.215 | 0.431 | 0.181 | 0.184 | 0.080 | **0.075** |
| | Brier | ↓ | 0.203 | 0.277 | 0.409 | 0.253 | 0.251 | 0.190 | **0.185** |
| *Sentence Level* | | | | | | | | | |
| TruthfulQA | acc@50 | ↑ | 0.314 | 0.267 | 0.233 | 0.405 | 0.341 | 0.415 | **0.430** |
| | cov@40 | ↑ | 0.136 | 0.005 | 0.224 | 0.500 | 0.332 | 0.500 | **0.510** |
| | ECE | ↓ | 0.138 | 0.323 | 0.510 | 0.060 | 0.148 | 0.087 | **0.058** |
| | Brier | ↓ | 0.218 | 0.349 | 0.474 | 0.194 | 0.190 | 0.193 | **0.176** |
| WikiQA | acc@50 | ↑ | 0.388 | 0.339 | 0.372 | 0.628 | 0.416 | 0.629 | **0.653** |
| | cov@50 | ↑ | 0.012 | 0.004 | 0.202 | **0.621** | 0.258 | 0.330 | 0.338 |
| | ECE | ↓ | 0.075 | 0.239 | 0.535 | 0.136 | 0.139 | 0.068 | **0.055** |
| | Brier | ↓ | 0.212 | 0.299 | 0.518 | 0.243 | 0.225 | 0.211 | **0.208** |
| **Average** | acc@50 | ↑ | 0.451 | 0.384 | 0.385 | 0.513 | 0.428 | 0.520 | **0.541** |
| | ECE | ↓ | 0.118 | 0.225 | 0.462 | 0.125 | 0.163 | 0.070 | **0.058** |
| | Brier | ↓ | 0.206 | 0.281 | 0.443 | 0.216 | 0.230 | 0.182 | **0.178** |

Table 3: Comparison of HADEMIF with hallucination detection and mitigation methods on the CAT benchmark for phrase- and sentence-level tasks. The results demonstrate that HADEMIF consistently outperforms other baselines across a wide range of tasks.

original LLM. Moreover, it decreases the ECE and Brier scores by 10% (from 0.107 to 0.096) and 6% (from 0.221 to 0.208), respectively, for paragraph-level tasks. Additionally, **HADEMIF outperforms all four calibration methods, achieving the lowest average ECE and Brier scores, along with the highest average acc@50**. Among the model calibration approaches, Label Smoothing performs poorly across nearly all tasks, even falling short of the original LLM. While Temperature Scaling and LITCAB demonstrate improved performance compared to Label Smoothing, their results remain far from optimal. Furthermore, since Calibration-Tuning cannot achieve fine-grained calibration of the prediction distribution, its performance is inferior to that of our approach.

**Comparison with Hallucination Detection and Mitigation Methods** As shown in Table 3, **HADEMIF consistently outperforms other hallucination detection and mitigation methods across both phrase- and sentence-level tasks**. It achieves the highest average acc@50 and the lowest average ECE and Brier scores, underscoring its effectiveness in estimating model confidence. Notably, compared to the best baselines, HADEMIF improves acc@50 by 5% (from 0.513 to 0.541) and reduces ECE and Brier scores by 54% (from 0.125 to 0.058) and 18% (from 0.216 to 0.178), respectively. Additionally, Verbalization and P(IK) perform poorly among the baselines, suggesting that although LLMs contain knowledge capable of revealing their hallucinations, effective approaches for knowledge modeling and extraction are necessary. Moreover, although R-Tuning fine-tuned LLMs, the confidence scores it generated are binary, making them less accurate compared to methods that use quantitative values to represent confidence, thus limiting their effectiveness. Additionally, Self-Consistency emerges as the best-performing baseline, aligned with our subsequent observation in Sec. 4.6 that consistency plays a more crucial role in hallucination detection compared to others. These findings position HADEMIF as a reliable method for hallucination detection and mitigation in LLMs as it comprehensively captures hallucinations in both the output and internal spaces, leveraging the modeled hallucinations for effective logit calibration.

**Results of Fine-Tuning LLMs** Fine-tuning LLMs with Label Smoothing does not yield satisfactory calibration results, suggesting that it is not well-suited for complex tasks such as LLM finetuning. Moreover, although other fine-tuning approaches, such as Calibration-Tuning and R-Tuning, demonstrate improved performance, their performance still falls short of ours, as they cannot achieve fine-grained calibration of the prediction distribution. Specifically, compared to the best fine-tuning baseline, HADEMIF increases acc@50 from 0.476 to 0.541 and reduces the ECE and Brier scores

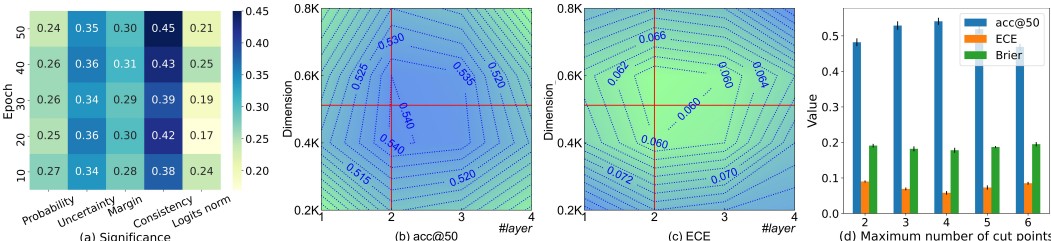

Figure 3: (a) Significance of prediction characteristics in hallucination detection measured by information gain throughout the training process. (b) and (c) Sensitivity analysis of the number and dimension of hidden layers in the MLP network. (d) Sensitivity analysis of the maximum number of cut points for characteristics in the D3T network.

from 0.075 to 0.058 and from 0.189 to 0.178, respectively, for both phrase- and sentence-level tasks. Additionally, it increases acc@50 from 0.597 to 0.621 and decreases ECE and Brier scores by 29% (from 0.136 to 0.096) and 12% (from 0.236 to 0.208), respectively, for paragraph-level tasks.

## 4.6 SIGNIFICANCE OF PREDICTION CHARACTERISTICS

Previous studies have employed various metrics, such as entropy and consistency, for hallucination detection (Xiao & Wang, 2021; Yang et al., 2023; Zhou et al., 2023). However, no consensus has been established regarding the most critical one for this task. The inherent interpretability of the D3T model facilitates a comprehensive analysis of the relative importance of different prediction characteristics in hallucination detection. Due to the feature splitting process in D3T, we can apply information gain to evaluate the significance of each metric. For the margin characteristic, we calculate the average information gain across both Top-1 and Top-$K$ margins. Fig. 3(a) illustrates the significance of various characteristics during the training process, leading to three key observations: **(1)** Consistency plays a more pivotal role than uncertainty in hallucination detection; **(2)** In addition to consistency and uncertainty, metrics such as margin, probability distribution, and logits norm have proven to be effective in detecting hallucinations in LLMs; and **(3)** A systematic approach that integrates multiple indicators is more effective than reliance on a single metric.

## 4.7 ABLATION STUDIES

We conduct ablation studies to evaluate the impact of the complexity of two hallucination detection networks on the effectiveness of our approach. This complexity is influenced by the number and dimensions of hidden layers in the MLP network, as well as the maximum number of cut points for the characteristics in the D3T model. We report the average performance across phrase- and sentence-level tasks. As illustrated in Figs. 3(b) and (c), performance remains stable with two or three hidden layers and layer dimensions ranging from 0.4K to 0.6K in the MLP network. Accordingly, we recommend employing an MLP network with two hidden layers, each with a dimension of 512. Furthermore, the model performs well when the maximum number of cut points in the D3T model is set to $\{3, 4, 5\}$, with peak performance observed at four cut points.

## 5 CONCLUSION

This study introduces HADEMIF, an innovative framework designed for detecting and mitigating hallucinations in LLMs. The framework utilizes two compact neural networks to identify hallucinations within both the output and internal spaces of LLMs. By calibrating model predictions based on the outputs of these networks, HADEMIF effectively reduces hallucinations and enhances the reliability of generated outputs. Our framework operates during both the inference and fine-tuning phases of LLMs by training two small-scale networks, which requires less than 2% additional parameters compared to the original LLMs. We evaluate the performance of the HADEMIF framework using the CAT benchmark, considering tasks with varying response lengths. The results demonstrate that HADEMIF significantly outperforms existing baselines in both hallucination detection and model calibration, highlighting its superior efficacy.

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

|  | NQ | SciQ | TriviaQA | TruthfulQA | WikiQA | BioGen | WikiGen |
|---|---|---|---|---|---|---|---|
| # Train | 2K | 2K | 2K | 397 | 1040 | 500 | 500 |
| # Test | 1K | 1K | 1K | 420 | 293 | 183 | 100 |
| Avg. # Response tokens | 2.2 | 1.5 | 2.3 | 35.5 | 25.7 | 77.3 | 104.0 |
| Max. # Response tokens | 6 | 12 | 42 | 24 | 166 | 318 | 197 |
| Min. # Response tokens | 1 | 1 | 1 | 1 | 1 | 21 | 33 |

Table 4: Task statistics in the CAT benchmark (Liu et al., 2024).

| Task | Question-Response Pair |
|---|---|
| ***Phrase Level*** | |
| NQ | *Question:* When did big air snowboarding become an olympic sport? *Response:* 2018 |
| SciQ | *Question:* In order to create food, what do photosynthetic protists use? *Response:* Light energy |
| TriviaQA | *Question:* What was Kevin Kline's first movie? *Response:* Sophie's Choice |
| ***Sentence Level*** | |
| TruthfulQA | *Question:* Which language do all people in Europe speak? *Response:* There's no single language that all Europeans speak |
| WikiQA | *Question:* How much is 1 tablespoon of water? *Response:* This tablespoon has a capacity of about 15 mL |
| ***Paragraph Level*** | |
| BioGen | *Question:* Write a paragraph for Bill Tobin's biography. *Response:* Ron Meagher (born October 2, 1941, Oakland, California, USA) is best known as the bassist of the American rock band The Beau Brummels. When guitarist-songwriter Ron Elliott was putting the... |
| WikiGen | *Question:* Write a paragraph about The Beatles. *Response:* The Beatles were an English rock band formed in Liverpool in 1960, comprising John Lennon, Paul McCartney, George Harrison, and Ringo Starr. They are regarded as the most influential band of all time... |

Table 5: Illustration of question-response pairs from tasks in the CAT benchmark.

# A APPENDIX

## A.1 MORE DETAILS OF THE CAT BENCHMARK

Table 4 summarizes the statistics of tasks from the CAT benchmark, while Table 5 provides examples of question-answer pairs in these tasks. We employ seven tasks from the CAT benchmark, following Liu et al. (2024), which include phrase-level tasks: NQ[6], SciQ[7], and TriviaQA[8]; sentence-level tasks: TruthfulQA[9] and WikiQA[10]; and paragraph-level tasks: BioGen[11] and WikiGen. For the three phrase-level tasks, 1K samples are used for testing and 2K samples for training. For TruthfulQA, which lacks an official training set, 397 instances are randomly sampled from the original test set for training and the remaining instances are utilized for testing. For the WikiQA dataset, the training set consists of 1,040 instances, while the test set contains 293 instances. For BioGen, a total of 683 names are compiled from (Min et al., 2023), of which 183 names are designated for evaluation and the remaining 500 are utilized for training. Similarly, for the WikiGen task, 600 entities

---

[6] https://github.com/google-research-datasets/natural-questions
[7] https://huggingface.co/datasets/allenai/sciq
[8] https://nlp.cs.washington.edu/triviaqa/
[9] https://github.com/sylinrl/TruthfulQA
[10] https://huggingface.co/datasets/microsoft/wiki_qa
[11] https://github.com/shmsw25/FActScore

| Task | Metric | | GPT-2 XL (1.5B) | | GPT-J (6B) | | Vicuna-13B | |
|------|--------|---|---|---|---|---|---|---|
| **_Phrase Level_** | | | | | | | | |
| NQ | acc@50 | ↑ | 0.062 | 0.130 | 0.146 | 0.227 | 0.246 | 0.323 |
| | cov@50 | ↑ | 0.001 | 0.076 | 0.057 | 0.100 | 0.113 | 0.124 |
| | ECE | ↓ | 0.045 | 0.033 | 0.059 | 0.024 | 0.204 | 0.045 |
| | Brier | ↓ | 0.055 | 0.016 | 0.079 | 0.043 | 0.224 | 0.135 |
| SciQ | acc@50 | ↑ | 0.258 | 0.274 | 0.620 | 0.665 | 0.678 | 0.723 |
| | cov@90 | ↑ | 0.007 | 0.019 | 0.135 | 0.157 | 0.142 | 0.166 |
| | ECE | ↓ | 0.059 | 0.026 | 0.133 | 0.087 | 0.244 | 0.187 |
| | Brier | ↓ | 0.137 | 0.126 | 0.209 | 0.166 | 0.318 | 0.204 |
| TriviaQA | acc@50 | ↑ | 0.100 | 0.187 | 0.270 | 0.294 | 0.464 | 0.463 |
| | cov@60 | ↑ | 0.000 | 0.135 | 0.128 | 0.245 | 0.268 | 0.372 |
| | ECE | ↓ | 0.063 | 0.028 | 0.068 | 0.031 | 0.186 | 0.121 |
| | Brier | ↓ | 0.069 | 0.066 | 0.115 | 0.110 | 0.238 | 0.230 |
| **Average** | acc@50 | ↑ | 0.140 | 0.197 | 0.345 | 0.395 | 0.463 | 0.503 |
| | ECE | ↓ | 0.056 | 0.029 | 0.087 | 0.047 | 0.211 | 0.118 |
| | Brier | ↓ | 0.087 | 0.069 | 0.134 | 0.106 | 0.260 | 0.190 |
| **_Sentence Level_** | | | | | | | | |
| TruthfulQA | acc@50 | ↑ | 0.186 | 0.290 | 0.162 | 0.273 | 0.552 | 0.595 |
| | cov@40 | ↑ | 0.005 | 0.198 | 0.040 | 0.230 | 0.998 | 0.998 |
| | ECE | ↓ | 0.041 | 0.036 | 0.112 | 0.044 | 0.200 | 0.088 |
| | Brier | ↓ | 0.118 | 0.071 | 0.136 | 0.090 | 0.303 | 0.205 |
| WikiQA | acc@50 | ↑ | 0.099 | 0.201 | 0.240 | 0.447 | 0.421 | 0.506 |
| | cov@50 | ↑ | 0.000 | 0.122 | 0.000 | 0.189 | 0.053 | 0.211 |
| | ECE | ↓ | 0.063 | 0.040 | 0.045 | 0.021 | 0.211 | 0.134 |
| | Brier | ↓ | 0.125 | 0.103 | 0.149 | 0.135 | 0.304 | 0.285 |
| **Average** | acc@50 | ↑ | 0.143 | 0.246 | 0.201 | 0.360 | 0.487 | 0.551 |
| | ECE | ↓ | 0.052 | 0.038 | 0.079 | 0.033 | 0.206 | 0.111 |
| | Brier | ↓ | 0.122 | 0.087 | 0.143 | 0.113 | 0.304 | 0.245 |
| **_Paragraph Level_** | | | | | | | | |
| BioGen | acc@50 | ↑ | – | – | 0.228 | 0.267 | 0.380 | 0.392 |
| | cov@40 | ↑ | – | – | 0.023 | 0.104 | 0.451 | 0.484 |
| | ECE | ↓ | – | – | 0.159 | 0.142 | 0.229 | 0.211 |
| | Brier | ↓ | – | – | 0.182 | 0.170 | 0.255 | 0.231 |
| WikiGen | acc@50 | ↑ | – | – | 0.395 | 0.403 | 0.822 | 0.835 |
| | cov@80 | ↑ | – | – | 0.001 | 0.022 | 0.675 | 0.712 |
| | ECE | ↓ | – | – | 0.102 | 0.091 | 0.168 | 0.132 |
| | Brier | ↓ | – | – | 0.220 | 0.205 | 0.227 | 0.211 |
| **Average** | acc@50 | ↑ | – | – | 0.312 | 0.335 | 0.601 | 0.614 |
| | ECE | ↓ | – | – | 0.131 | 0.117 | 0.199 | 0.172 |
| | Brier | ↓ | – | – | 0.201 | 0.188 | 0.241 | 0.221 |

Table 6: Comparison of the HADEMIF approach (with fine-tuning) and baseline performance across GPT-2 XL (1.5B), GPT-J (6B), and Vicuna-13B on the CAT benchmark. Results for GPT-2 XL in paragraph-level tasks are excluded as the prompt length exceeds its context window limit. Scores where our approach surpasses the original LLM are highlighted in green . The proposed HADEMIF approach consistently exhibits superior performance compared to baseline results across multiple LLMs.

are randomly selected from the FEVER[12] dataset, each linked to a specific Wikipedia passage. Of these, 100 entities were set aside for evaluation, while the remaining 500 were utilized for training.

---

[12]https://github.com/awslabs/fever

| Task | Metric | | LLaMA-7B | | LLaMA-30B | | Llama2-7B | | Llama2-13B | | Llama3-8B | |
|---|---|---|---|---|---|---|---|---|---|---|---|---|
| ***Phrase Level*** | | | | | | | | | | | | |
| NQ | acc@50 | ↑ | 0.358 | 0.389 | 0.466 | 0.512 | 0.288 | 0.355 | 0.448 | 0.491 | 0.510 | 0.542 |
| | cov@50 | ↑ | 0.271 | 0.282 | 0.445 | 0.503 | 0.115 | 0.120 | 0.407 | 0.415 | 0.502 | 0.516 |
| | ECE | ↓ | 0.144 | 0.032 | 0.169 | 0.037 | 0.171 | 0.026 | 0.139 | 0.019 | 0.126 | 0.020 |
| | Brier | ↓ | 0.174 | 0.103 | 0.192 | 0.112 | 0.196 | 0.119 | 0.187 | 0.100 | 0.179 | 0.097 |
| SciQ | acc@50 | ↑ | 0.756 | 0.778 | 0.874 | 0.899 | 0.764 | 0.766 | 0.844 | 0.842 | 0.881 | 0.892 |
| | cov@90 | ↑ | 0.261 | 0.278 | 0.423 | 0.435 | 0.211 | 0.228 | 0.375 | 0.405 | 0.435 | 0.459 |
| | ECE | ↓ | 0.126 | 0.081 | 0.107 | 0.065 | 0.094 | 0.076 | 0.102 | 0.066 | 0.100 | 0.061 |
| | Brier | ↓ | 0.210 | 0.200 | 0.186 | 0.157 | 0.203 | 0.200 | 0.197 | 0.154 | 0.174 | 0.139 |
| TriviaQA | acc@50 | ↑ | 0.474 | 0.493 | 0.462 | 0.469 | 0.500 | 0.501 | 0.454 | 0.486 | 0.582 | 0.620 |
| | cov@50 | ↑ | 0.029 | 0.157 | 0.156 | 0.264 | 0.111 | 0.240 | 0.169 | 0.270 | 0.336 | 0.369 |
| | ECE | ↓ | 0.137 | 0.078 | 0.052 | 0.036 | 0.112 | 0.075 | 0.087 | 0.054 | 0.061 | 0.040 |
| | Brier | ↓ | 0.213 | 0.197 | 0.174 | 0.158 | 0.203 | 0.185 | 0.190 | 0.181 | 0.146 | 0.123 |
| **Average** | acc@50 | ↑ | 0.529 | 0.553 | 0.601 | 0.627 | 0.517 | 0.541 | 0.582 | 0.606 | 0.658 | 0.685 |
| | ECE | ↓ | 0.136 | 0.064 | 0.109 | 0.046 | 0.126 | 0.059 | 0.109 | 0.046 | 0.096 | 0.040 |
| | Brier | ↓ | 0.199 | 0.167 | 0.184 | 0.142 | 0.201 | 0.168 | 0.191 | 0.145 | 0.166 | 0.120 |
| ***Sentence Level*** | | | | | | | | | | | | |
| TruthfulQA | acc@50 | ↑ | 0.012 | 0.145 | 0.433 | 0.542 | 0.314 | 0.430 | 0.362 | 0.441 | 0.571 | 0.590 |
| | cov@40 | ↑ | 0.117 | 0.419 | 0.648 | 0.697 | 0.136 | 0.510 | 0.350 | 0.613 | 0.698 | 0.721 |
| | ECE | ↓ | 0.120 | 0.050 | 0.110 | 0.043 | 0.138 | 0.058 | 0.132 | 0.060 | 0.088 | 0.046 |
| | Brier | ↓ | 0.184 | 0.143 | 0.235 | 0.193 | 0.218 | 0.176 | 0.233 | 0.180 | 0.187 | 0.145 |
| WikiQA | acc@50 | ↑ | 0.322 | 0.608 | 0.264 | 0.462 | 0.388 | 0.653 | 0.455 | 0.655 | 0.489 | 0.620 |
| | cov@50 | ↑ | 0.086 | 0.341 | 0.078 | 0.218 | 0.012 | 0.338 | 0.358 | 0.434 | 0.430 | 0.458 |
| | ECE | ↓ | 0.108 | 0.066 | 0.142 | 0.103 | 0.075 | 0.055 | 0.064 | 0.037 | 0.077 | 0.052 |
| | Brier | ↓ | 0.190 | 0.176 | 0.182 | 0.173 | 0.212 | 0.208 | 0.192 | 0.175 | 0.168 | 0.154 |
| **Average** | acc@50 | ↑ | 0.167 | 0.377 | 0.349 | 0.502 | 0.351 | 0.542 | 0.409 | 0.548 | 0.530 | 0.605 |
| | ECE | ↓ | 0.114 | 0.058 | 0.126 | 0.073 | 0.107 | 0.057 | 0.098 | 0.049 | 0.083 | 0.049 |
| | Brier | ↓ | 0.187 | 0.160 | 0.209 | 0.183 | 0.215 | 0.192 | 0.213 | 0.178 | 0.178 | 0.150 |
| ***Paragraph Level*** | | | | | | | | | | | | |
| BioGen | acc@50 | ↑ | 0.220 | 0.250 | 0.313 | 0.331 | 0.347 | 0.358 | 0.485 | 0.500 | 0.513 | 0.532 |
| | cov@40 | ↑ | 0.134 | 0.218 | 0.300 | 0.398 | 0.066 | 0.160 | 0.999 | 0.998 | 0.457 | 0.503 |
| | ECE | ↓ | 0.143 | 0.131 | 0.105 | 0.080 | 0.169 | 0.160 | 0.114 | 0.068 | 0.097 | 0.061 |
| | Brier | ↓ | 0.153 | 0.139 | 0.173 | 0.158 | 0.269 | 0.255 | 0.260 | 0.244 | 0.199 | 0.161 |
| WikiGen | acc@50 | ↑ | 0.667 | 0.685 | 0.750 | 0.762 | 0.876 | 0.884 | 0.900 | 0.911 | 0.925 | 0.931 |
| | cov@80 | ↑ | 0.220 | 0.250 | 0.354 | 0.382 | 0.745 | 0.774 | 0.914 | 0.935 | 0.875 | 0.924 |
| | ECE | ↓ | 0.102 | 0.087 | 0.165 | 0.154 | 0.045 | 0.032 | 0.048 | 0.031 | 0.050 | 0.044 |
| | Brier | ↓ | 0.239 | 0.200 | 0.252 | 0.238 | 0.172 | 0.161 | 0.164 | 0.147 | 0.169 | 0.148 |
| **Average** | acc@50 | ↑ | 0.444 | 0.468 | 0.532 | 0.547 | 0.612 | 0.621 | 0.693 | 0.706 | 0.719 | 0.732 |
| | ECE | ↓ | 0.123 | 0.109 | 0.135 | 0.117 | 0.107 | 0.096 | 0.081 | 0.050 | 0.074 | 0.053 |
| | Brier | ↓ | 0.196 | 0.170 | 0.213 | 0.198 | 0.221 | 0.208 | 0.212 | 0.196 | 0.184 | 0.155 |

Table 7: Comparison of the HADEMIF approach (with fine-tuning) and baseline performance across LLaMA, Llama2, and Llama3 models on the CAT benchmark. The HADEMIF framework consistently achieves improvements over baseline performance across multiple LLMs.

## A.2 MORE EXPERIMENTAL SETTINGS

Besides Llama2-7B, we assess the performance of our approach utilizing seven other LLMs: GPT-2 XL (1.5B) (Radford et al., 2019), GPT-J (6B) (Wang & Komatsuzaki, 2021), LLaMA-7B (Touvron et al., 2023a), LLaMA-30B, Llama2-13B, Vicuna-13B (Chiang et al., 2023), and Llama3-8B (Dubey et al., 2024). Moreover, to ensure consistency across tasks, we employ ICL, as not all LLMs demonstrate strong zero-shot performance. The settings follow those described in (Liu et al., 2024). Specifically, for phrase- and sentence-level tasks, we utilize fifteen demonstrations. For the two paragraph-level tasks, we employ five examples per task, accounting for the extended length of the demonstrations. The queries for BioGen and WikiGen are phrased as "Write a paragraph for [Name]'s biography" and "Write a paragraph about [Entity]," respectively. Regarding the initialization of the two hallucination detection networks, the MLP network is initialized using He initialization He et al. (2015), as this method is effective for layers with ReLU activation function. For the D3T model, following the initialization approach in DNDT (Yang et al., 2018), all parameters are initialized using Xavier initialization Glorot & Bengio (2010) with a uniform distribution.

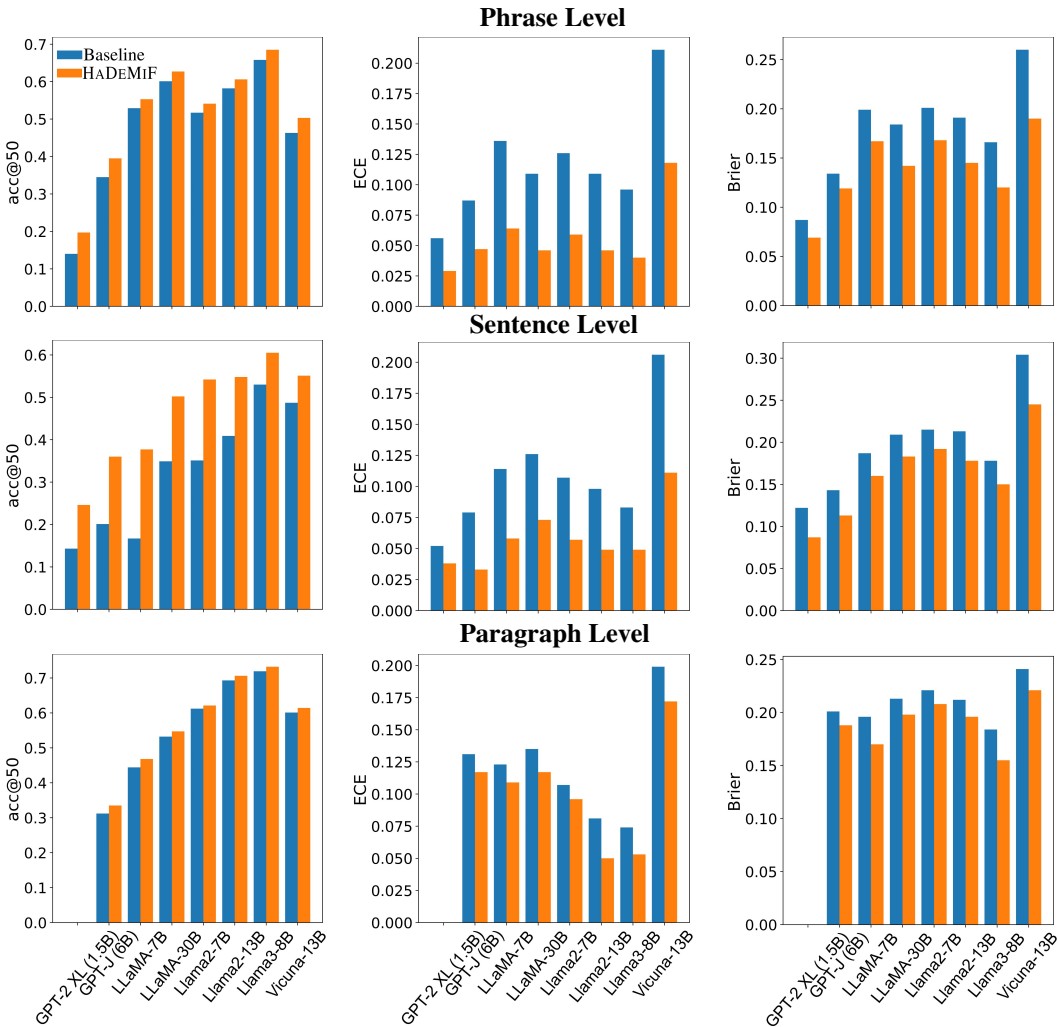

Figure 4: Bar charts illustrating the averaged acc@50, ECE, and Brier scores for eight widely used LLMs as assessed on the CAT benchmark.

## A.3 PERFORMANCE COMPARISON ON THE CAT BENCHMARK USING VARIOUS LLMS

Table 6 presents a comparative analysis of HADEMIF against baseline performance across GPT-2 XL (1.5B)[13], GPT-J (6B)[14], and Vicuna-13B[15], while Table 7 provides comparison results for the LLaMA, Llama2, and Llama3 models, including LLaMA-7B[16], LLaMA-30B[17], Llama2-7B[18], Llama2-13B[19], and Llama3-8B[20]. HADEMIF consistently outperforms the original LLMs in generation tasks on the CAT benchmark. These results highlight the effectiveness of our approach in hallucination detection and model calibration. Moreover, Fig. 4 presents a performance comparison between HADEMIF and the original LLMs across various evaluation metrics. HADEMIF consistently outperforms the original performance of several LLMs. Notably, while the GPT-2 model exhibits lower accuracy, it demonstrates reduced overconfidence, as indicated by its smaller ECE

---

[13] https://huggingface.co/openai-community/gpt2-xl

[14] https://huggingface.co/EleutherAI/gpt-j-6b

[15] https://lmsys.org/blog/2023-03-30-vicuna/

[16] https://huggingface.co/huggyllama/llama-7b

[17] https://huggingface.co/huggyllama/llama-30b

[18] https://huggingface.co/meta-llama/Llama-2-7b

[19] https://huggingface.co/meta-llama/Llama-2-13b-hf

[20] https://huggingface.co/meta-llama/Meta-Llama-3-8B

| Dataset | NQ | | | TrivalQA | | |
|---|---|---|---|---|---|---|
| Metric | AUC$_s$ ↑ | AUC$_r$ ↑ | PCC ↑ | AUC$_s$ ↑ | AUC$_r$ ↑ | PCC ↑ |
| Perplexity | 0.740 | 0.747 | 0.301 | 0.836 | 0.836 | 0.544 |
| LN-Entropy | 0.728 | 0.737 | 0.298 | 0.834 | 0.832 | 0.540 |
| Lexical Similarity | 0.738 | 0.759 | 0.306 | 0.826 | 0.840 | 0.556 |
| HADEMIF | **0.772** | **0.776** | **0.389** | **0.843** | **0.842** | **0.581** |

Table 8: Evaluation of hallucination detection performance across different methods on the NQ and TriviaQA tasks. The AUC$_s$ represents the AUROC score using sentence similarity as the measure of correctness, while AUC$_r$ represents the AUROC score using ROUGE-L as the correctness measure. HADEMIF consistently surpasses previous methods that rely on a single aspect of metrics, such as uncertainty and consistency.

| Method | acc@50 ↑ | cov@50 ↑ | ECE ↓ | Brier ↓ |
|---|---|---|---|---|
| Original LLM | 0.288 | 0.115 | 0.171 | 0.196 |
| Label Smoothing | 0.208 | 0.061 | 0.186 | 0.212 |
| Temp. Scaling | 0.288 | 0.115 | 0.165 | 0.193 |
| LITCAB | 0.300 | 0.105 | 0.101 | 0.169 |
| LITCAB w/ Temp. Scaling | 0.300 | 0.105 | 0.083 | 0.164 |
| HADEMIF (Rank 8) | **0.355** | 0.120 | **0.026** | **0.119** |
| HADEMIF (Rank 16) | 0.350 | 0.116 | 0.030 | 0.121 |
| HADEMIF (Rank 32) | 0.352 | 0.125 | 0.029 | 0.120 |

Table 9: Applicability with increasing training complexity.

and Brier scores. Additionally, Vicuna-13B, which has been fine-tuned from LLaMA-13B using user-shared conversations, shows inferior performance relative to LLaMA, as indicated by its higher average ECE and Brier scores.

## A.4 COMPARISON WITH MORE HALLUCINATION DETECTION METHODS

We compare HADEMIF with three other hallucination detection approaches on the NQ and TriviaQA datasets using the LLaMA-7B model. The baseline methods include uncertainty-based approaches, such as Perplexity (Ren et al., 2022; Chen et al., 2025; Du et al., 2022) and Length-Normalized Entropy (LN-Entropy) (Malinin & Gales, 2020), as well as the consistency-based metric, Lexical Similarity (Lin et al., 2022). Following previous studies (Lin et al., 2024; Ren et al., 2022; Chen et al., 2024a), we use the area under the receiver operating characteristic curve (AUROC) and the Pearson Correlation Coefficient (PCC) as performance metrics. For our approach, the hallucination detection score is derived from the D3T model. The results for all methods are presented in Table 8. Our proposed HADEMIF framework outperforms previous methods that rely solely on a single aspect of characteristics, thanks to its incorporation of a comprehensive set of prediction characteristics for hallucination detection.

## A.5 APPLICABILITY WITH INCREASING TRAINING COMPLEXITY

To evaluate the applicability of our approach as training complexity increases, we examine how performance varies with the rank in LoRA (Hu et al., 2022). A larger rank indicates that more parameters are trainable. The results on the NQ dataset are presented in Table 9. The performance of HADEMIF consistently surpasses that of the comparison methods across different levels of training complexity. We also observe that a relatively small rank can already yield excellent results, which aligns with findings from the original LoRA paper.

| Dataset | Metric | | HADEMIF with D3T | HADEMIF with DNDT |
|---|---|---|---|---|
| NQ | acc@50 | ↑ | **0.315** | 0.311 |
| | cov@50 | ↑ | **0.115** | 0.112 |
| | ECE | ↓ | **0.034** | 0.045 |
| | Brier | ↓ | **0.116** | 0.131 |
| SciQ | acc@50 | ↑ | **0.760** | 0.754 |
| | cov@90 | ↑ | **0.230** | 0.225 |
| | ECE | ↓ | **0.083** | 0.084 |
| | Brier | ↓ | **0.201** | 0.203 |

Table 10: Comparison of D3T and DNDT as hallucination detection networks for the output space. Utilizing D3T as the hallucination detection network consistently outperforms DNDT.

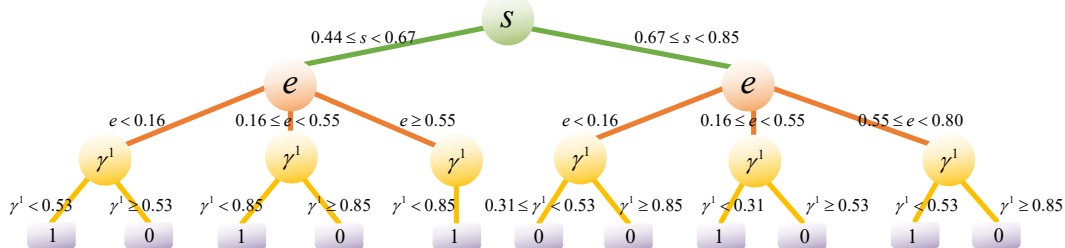

Figure 5: Visualization of a segment from the first three layers of the decision tree generated by the D3T model at the final training epoch for the NQ dataset. The outcomes at the leaf nodes reflect the majority classification results for their respective branches, with 1 denoting hallucination and 0 denoting non-hallucination. In this context, $s$, $e$, and $\gamma^1$ denote the consistency, uncertainty, and Top-1 margins, respectively.

## A.6 COMPARISON BETWEEN D3T AND DNDT

Unlike the DNDT model, which employs a fixed structure that results in numerous redundant branches, our D3T model employs a more flexible architecture that dynamically adjusts the number of cut bins for each input feature throughout the training process. To demonstrate the superiority of D3T over DNDT, we conduct experiments on both the NQ and SciQ datasets, employing D3T or DNDT as the hallucination detection networks for the output space. In these experiments, the maximum number of cut points for D3T and the fixed number of cut points for DNDT are both set to 4. The results presented in Table 10 show that D3T consistently outperforms DNDT as the hallucination detection network. The primary reason is that the adaptability of D3T enables it to better align with the local characteristics of the data, thereby reducing the risk of overfitting.

## A.7 DECISION RULES AND GENERATED DECISION TREE

The interpretability of the D3T model enables the visualization of decision rules, providing insights into how combinations of predictive characteristics determine the presence of hallucinations. In Table 11, we present a total of fifty decision rules learned by D3T. The first column represents the combinations of characteristics that satisfy various conditions, while the second column displays the corresponding hallucination detection outcomes for each combination. By analyzing these decision rules, we find that low consistency, high uncertainty, small margins, and low maximum probability are typically indicative of hallucinations, which aligns with prevailing perceptions. However, the decision rules also reveal that this relationship is not absolute. Therefore, relying solely on a single metric, as previous methods have done, is inadequate for effectively detecting hallucinations.

Moreover, to more intuitively reflect the model's decision-making, we visualize the decision tree, where a hierarchical arrangement of prediction characteristics is established based on their information gains, similar to traditional decision trees. Specifically, the characteristics are organized in layers based on their information gains, which reflect their importance (calculated in Sec. 4.6), ar-

| Combination of characteristics | Decision |
|---|---|
| $s < 0.23$ & $e \geq 0.80$ & $0.31 \leq \gamma^1 < 0.53$ & $\gamma^K < 0.38$ & $0.29 \leq p^m < 0.61$ & $|\boldsymbol{u}| < 0.25$ | 1 |
| $s < 0.23$ & $0.55 \leq e < 0.80$ & $\gamma^1 < 0.31$ & $0.38 \leq \gamma^K < 0.67$ & $0.29 \leq p^m < 0.61$ & $0.25 \leq |\boldsymbol{u}| < 0.44$ | 1 |
| $s < 0.23$ & $0.55 \leq e < 0.80$ & $0.31 \leq \gamma^1 < 0.53$ & $0.38 \leq \gamma^K < 0.67$ & $0.29 \leq p^m < 0.61$ & $0.25 \leq |\boldsymbol{u}| < 0.44$ | 1 |
| $0.23 \leq s < 0.44$ & $e \geq 0.80$ & $0.31 \leq \gamma^1 < 0.53$ & $\gamma^K < 0.38$ & $0.29 \leq p^m < 0.61$ & $0.25 \leq |\boldsymbol{u}| < 0.44$ | 1 |
| $0.23 \leq s < 0.44$ & $0.55 \leq e < 0.80$ & $\gamma^1 < 0.31$ & $0.38 \leq \gamma^K < 0.67$ & $p^m < 0.29$ & $|\boldsymbol{u}| < 0.25$ | 1 |
| $0.23 \leq s < 0.44$ & $0.16 \leq e < 0.55$ & $\gamma^1 < 0.31$ & $\gamma^K < 0.38$ & $p^m < 0.29$ & $0.25 \leq |\boldsymbol{u}| < 0.44$ | 1 |
| $0.44 \leq s < 0.67$ & $e < 0.16$ & $\gamma^1 < 0.31$ & $\gamma^K < 0.38$ & $p^m < 0.29$ & $|\boldsymbol{u}| < 0.25$ | 1 |
| $0.44 \leq s < 0.67$ & $e < 0.16$ & $\gamma^1 < 0.31$ & $0.38 \leq \gamma^K < 0.67$ & $p^m < 0.29$ & $0.25 \leq \boldsymbol{u} < 0.44$ | 1 |
| $0.44 \leq s < 0.67$ & $e < 0.16$ & $0.31 \leq \gamma^1 < 0.53$ & $\gamma^K \leq 0.38$ & $0.29 \leq p^m < 0.61$ & $|\boldsymbol{u}| < 0.25$ | 1 |
| $0.44 \leq s < 0.67$ & $e < 0.16$ & $0.53 \leq \gamma^1 < 0.85$ & $\gamma^K \geq 0.88$ & $0.61 \leq p^m < 0.84$ & $|\boldsymbol{u}| \geq 0.65$ | 0 |
| $0.44 \leq s < 0.67$ & $e < 0.16$ & $0.53 \leq \gamma^1 < 0.85$ & $\gamma^K \geq 0.88$ & $p^m \geq 0.84$ & $|\boldsymbol{u}| \geq 0.65$ | 0 |
| $0.44 \leq s < 0.67$ & $e < 0.16$ & $\gamma^1 \geq 0.85$ & $\gamma^K \geq 0.88$ & $p^m \geq 0.84$ & $0.44 \leq |\boldsymbol{u}| < 0.65$ | 0 |
| $0.44 \leq s < 0.67$ & $0.16 \leq e < 0.55$ & $\gamma^1 < 0.31$ & $\gamma^K < 0.38$ & $p^m < 0.29$ & $0.25 \leq |\boldsymbol{u}| < 0.44$ | 1 |
| $0.44 \leq s < 0.67$ & $0.16 \leq e < 0.55$ & $0.31 \leq \gamma^1 < 0.53$ & $\gamma^K < 0.38$ & $0.29 \leq p^m < 0.61$ & $|\boldsymbol{u}| < 0.25$ | 1 |
| $0.44 \leq s < 0.67$ & $0.16 \leq e < 0.55$ & $\gamma^1 < 0.31$ & $0.38 \leq \gamma^K < 0.67$ & $p^m < 0.29$ & $|\boldsymbol{u}| < 0.25$ | 1 |
| $0.44 \leq s < 0.67$ & $0.16 \leq e < 0.55$ & $0.31 \leq \gamma^1 < 0.53$ & $\gamma^K < 0.38$ & $0.29 \leq p^m < 0.61$ & $0.25 \leq |\boldsymbol{u}| < 0.44$ | 1 |
| $0.44 \leq s < 0.67$ & $0.16 \leq e < 0.55$ & $0.53 \leq \gamma^1 < 0.85$ & $\gamma^K < 0.38$ & $0.29 \leq p^m < 0.61$ & $|\boldsymbol{u}| < 0.25$ | 1 |
| $0.44 \leq s < 0.67$ & $0.16 \leq e < 0.55$ & $\gamma^1 \geq 0.85$ & $\gamma^K > 0.88$ & $p^m \geq 0.84$ & $0.44 \leq |\boldsymbol{u}| < 0.65$ | 0 |
| $0.44 \leq s < 0.67$ & $0.16 \leq e < 0.55$ & $\gamma^1 \geq 0.85$ & $0.67 \leq \gamma^K < 0.88$ & $p^m \geq 0.84$ & $|\boldsymbol{u}| \geq 0.65$ | 0 |
| $0.44 \leq s < 0.67$ & $0.16 \leq e < 0.55$ & $\gamma^1 \geq 0.85$ & $0.67 \leq \gamma^K < 0.88$ & $0.61 \leq p^m < 0.84$ & $|\boldsymbol{u}| \geq 0.65$ | 0 |
| $0.44 \leq s < 0.67$ & $0.55 \leq e < 0.80$ & $\gamma^1 < 0.31$ & $\gamma^K < 0.38$ & $0.29 \leq p^m < 0.61$ & $|\boldsymbol{u}| < 0.25$ | 1 |
| $0.44 \leq s < 0.67$ & $0.55 \leq e < 0.80$ & $0.31 \leq \gamma^1 < 0.53$ & $\gamma^K < 0.38$ & $0.29 \leq p^m < 0.61$ & $|\boldsymbol{u}| < 0.25$ | 1 |
| $0.44 \leq s < 0.67$ & $0.55 \leq e < 0.80$ & $0.31 \leq \gamma^1 < 0.53$ & $\gamma^K < 0.38$ & $0.29 \leq p^m < 0.61$ & $0.25 \leq |\boldsymbol{u}| < 0.44$ | 1 |
| $0.44 \leq s < 0.67$ & $0.55 \leq e < 0.80$ & $0.53 \leq \gamma^1 < 0.85$ & $\gamma^K < 0.38$ & $0.61 \leq p^m < 0.84$ & $|\boldsymbol{u}| < 0.25$ | 1 |
| $0.44 \leq s < 0.67$ & $0.55 \leq e < 0.80$ & $0.31 \leq \gamma^1 < 0.53$ & $\gamma^K < 0.38$ & $0.29 \leq p^m < 0.61$ & $0.44 \leq |\boldsymbol{u}| < 0.65$ | 1 |
| $0.67 \leq s < 0.85$ & $e < 0.16$ & $0.31 \leq \gamma^1 < 0.53$ & $\gamma^K \geq 0.88$ & $p^m \geq 0.84$ & $|\boldsymbol{u}| \geq 0.65$ | 0 |
| $0.67 \leq s < 0.85$ & $e < 0.16$ & $0.53 \leq \gamma^1 < 0.85$ & $\gamma^K \geq 0.88$ & $0.61 \leq p^m < 0.84$ & $0.44 \leq |\boldsymbol{u}| < 0.65$ | 0 |
| $0.67 \leq s < 0.85$ & $e < 0.16$ & $0.53 \leq \gamma^1 < 0.85$ & $0.67 \leq \gamma^K < 0.88$ & $0.61 \leq p^m < 0.84$ & $|\boldsymbol{u}| \geq 0.65$ | 0 |
| $0.67 \leq s < 0.85$ & $e < 0.16$ & $0.53 \leq \gamma^1 < 0.85$ & $\gamma^K < 0.38$ & $0.61 \leq p^m < 0.84$ & $|\boldsymbol{u}| \geq 0.65$ | 0 |
| $0.67 \leq s < 0.85$ & $e < 0.16$ & $0.53 \leq \gamma^1 < 0.85$ & $\gamma^K < 0.38$ & $0.61 \leq p^m < 0.84$ & $0.44 \leq |\boldsymbol{u}| < 0.65$ | 0 |
| $0.67 \leq s < 0.85$ & $e < 0.16$ & $0.53 \leq \gamma^1 < 0.85$ & $\gamma^K < 0.38$ & $0.61 \leq p^m < 0.84$ & $0.25 \leq |\boldsymbol{u}| < 0.44$ | 0 |
| $0.67 \leq s < 0.85$ & $e < 0.16$ & $\gamma^1 \geq 0.85$ & $\gamma^K < 0.38$ & $p^m \geq 0.84$ & $|\boldsymbol{u}| \geq 0.65$ | 0 |
| $0.67 \leq s < 0.85$ & $e < 0.16$ & $\gamma^1 \geq 0.85$ & $\gamma^K \geq 0.88$ & $p^m \geq 0.84$ & $|\boldsymbol{u}| \geq 0.65$ | 0 |
| $0.67 \leq s < 0.85$ & $e < 0.16$ & $\gamma^1 \geq 0.85$ & $\gamma^K < 0.38$ & $p^m \geq 0.84$ & $0.44 \leq |\boldsymbol{u}| < 0.65$ | 0 |
| $0.67 \leq s < 0.85$ & $0.16 \leq e < 0.55$ & $\gamma^1 < 0.31$ & $\gamma^K < 0.38$ & $p^m < 0.29$ & $0.25 \leq |\boldsymbol{u}| < 0.44$ | 1 |
| $0.67 \leq s < 0.85$ & $0.16 \leq e < 0.55$ & $\gamma^1 < 0.31$ & $0.38 \leq \gamma^K < 0.67$ & $p^m < 0.29$ & $0.25 \leq |\boldsymbol{u}| < 0.44$ | 1 |
| $0.67 \leq s < 0.85$ & $0.16 \leq e < 0.55$ & $\gamma^1 < 0.31$ & $\gamma^K < 0.38$ & $0.29 \leq p^m < 0.61$ & $|\boldsymbol{u}| < 0.25$ | 1 |
| $0.67 \leq s < 0.85$ & $0.16 \leq e < 0.55$ & $0.53 \leq \gamma^1 < 0.85$ & $0.67 \leq \gamma^K < 0.88$ & $p^m \geq 0.84$ & $0.44 \leq |\boldsymbol{u}| < 0.65$ | 0 |
| $0.67 \leq s < 0.85$ & $0.16 \leq e < 0.55$ & $0.53 \leq \gamma^1 < 0.85$ & $\gamma^K \geq 0.88$ & $p^m \geq 0.84$ & $|\boldsymbol{u}| \geq 0.65$ | 0 |
| $0.67 \leq s < 0.85$ & $0.16 \leq e < 0.55$ & $\gamma^1 \geq 0.85$ & $0.67 \leq \gamma^K < 0.88$ & $p^m \geq 0.84$ & $0.44 \leq |\boldsymbol{u}| < 0.65$ | 0 |
| $0.67 \leq s < 0.85$ & $0.55 \leq e < 0.80$ & $0.31 \leq \gamma^1 < 0.53$ & $\gamma^K < 0.38$ & $0.29 \leq p^m < 0.61$ & $|\boldsymbol{u}| < 0.25$ | 1 |
| $0.67 \leq s < 0.85$ & $0.55 \leq e < 0.80$ & $0.31 \leq \gamma^1 < 0.53$ & $\gamma^K < 0.38$ & $0.29 \leq p^m < 0.61$ & $0.25 \leq |\boldsymbol{u}| < 0.44$ | 1 |
| $0.67 \leq s < 0.85$ & $0.55 \leq e < 0.80$ & $\gamma^1 < 0.31$ & $0.38 \leq \gamma^K < 0.67$ & $p^m < 0.29$ & $|\boldsymbol{u}| < 0.25$ | 1 |
| $0.67 \leq s < 0.85$ & $0.55 \leq e < 0.80$ & $\gamma^1 < 0.31$ & $0.38 \leq \gamma^K < 0.67$ & $0.29 \leq p^m < 0.61$ & $0.25 \leq |\boldsymbol{u}| < 0.44$ | 1 |
| $0.67 \leq s < 0.85$ & $0.55 \leq e < 0.80$ & $\gamma^1 \geq 0.85$ & $0.67 \leq \gamma^K < 0.88$ & $p^m \geq 0.84$ & $|\boldsymbol{u}| \geq 0.65$ | 0 |
| $0.67 \leq s < 0.85$ & $0.55 \leq e < 0.80$ & $\gamma^1 \geq 0.85$ & $\gamma^K \geq 0.88$ & $p^m \geq 0.84$ & $|\boldsymbol{u}| \geq 0.65$ | 0 |
| $s \geq 0.85$ & $e < 0.16$ & $\gamma^1 \geq 0.85$ & $0.38 \leq \gamma^K < 0.67$ & $p^m \geq 0.84$ & $0.44 \leq |\boldsymbol{u}| < 0.65$ | 0 |
| $s \geq 0.85$ & $0.16 \leq e < 0.55$ & $0.53 \leq \gamma^1 < 0.85$ & $\gamma^K \geq 0.88$ & $p^m \geq 0.84$ & $|\boldsymbol{u}| \geq 0.65$ | 0 |
| $s \geq 0.85$ & $0.16 \leq e < 0.55$ & $\gamma^1 \geq 0.85$ & $0.67 \leq \gamma^K < 0.88$ & $p^m \geq 0.84$ & $|\boldsymbol{u}| \geq 0.65$ | 0 |
| $s \geq 0.85$ & $e \geq 0.80$ & $\gamma^1 \geq 0.85$ & $\gamma^K \geq 0.88$ & $p^m \geq 0.84$ & $|\boldsymbol{u}| \geq 0.65$ | 0 |

Table 11: Examples of decision rules learned by D3T. We highlight the six most informative characteristic values, where $s$, $e$, $\gamma^1$, $\gamma^K$, $p^m$, and $|\boldsymbol{u}|$ represent consistency, uncertainty, top-1 margin, top-$K$ margin, maximum probability, and logits norm, respectively.

ranged in descending order. Fig. 5 presents a segment of the first three layers of the decision tree from the 50th epoch, where each path within the decision tree represents a decision rule. From this tree, we can find that consistency emerges as the most critical characteristic, followed by uncertainty and margin. Moreover, hallucinated generations typically exhibit low consistency, high uncertainty, and small margins, although the relationship between each metric and hallucination is not absolute.

## A.8 ADDITIONAL ABLATION STUDIES

We conduct additional ablation studies on the proposed HADEMIF framework, focusing on the complexity of the MLP network, the specific layer from which the hidden states are extracted, and the value of the temperature factor $\tau$. The average performance across both phrase- and sentence-

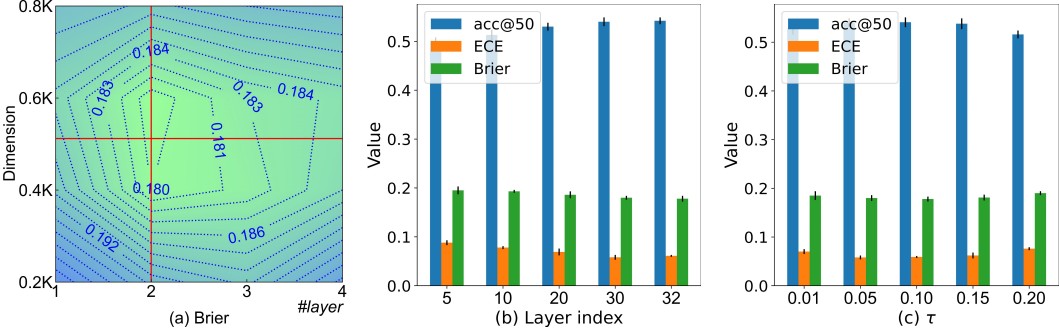

Figure 6: (a) Brier scores associated with varying numbers and dimensions of hidden layers in the MLP network. (b) Calibration performance based on internal states extracted from different layers. (c) Calibration performance based on varying values of $\tau$ in Eq. (1).

level tasks is reported. Fig. 6(a) demonstrates that model performance, measured by the Brier score, is stable when the number of hidden layers is selected from $\{2, 3\}$ and the dimension of the hidden layers ranges between 0.4K and 0.6K. Therefore, as mentioned in the main text, we recommend setting the number of layers to two and the dimension to 512. Additionally, Fig. 6(b) presents the calibration performance based on hidden states extracted from different layers. The results indicate that the internal states extracted from the later and middle layers yield superior performance compared to those from the earlier layers. Therefore, we directly utilize the hidden states preceding the logits. Fig.6(c) illustrates that the model performance remains stable when $\tau \in [0.05, 0.15]$. Therefore, we recommend setting $\tau$ to 0.1 for practical applications.

### A.9 Initialization for Hallucination Detection Networks

To identify the optimal initialization settings for the two hallucination detection networks, we compare four different initialization configurations under the condition of the same number of training iterations (40 epochs). The MLP network considers two initialization methods: He initialization (He et al., 2015) and Xavier initialization Glorot & Bengio (2010) with a uniform distribution (denoted as X-u). Moreover, D3T considers two initialization methods: X-u and Xavier initialization with a normal distribution (denoted as X-n). The comparison results are shown in Table 12. He initialization, specifically designed for ReLU activation functions, proves to be more advantageous for initializing the MLP network. For the D3T network, the performance difference between Xavier initialization with uniform and normal distributions is minimal. Consequently, for the MLP network, we utilize He initialization, while for the D3T model, all parameters are initialized using Xavier initialization with a uniform distribution.

### A.10 Limitations and Future Work

Our study effectively mitigates hallucinations of LLMs and enhances model calibration. However, there are several limitations. A key limitation is that the proposed method relies on internal information from LLMs, rendering it inapplicable to black-box models where users lack access to hidden states. Moreover, due to the introduction of new modules and the computation of additional metrics, our approach increases computational costs, as analyzed in Sec. A.12. Since the majority of the increased time is attributable to the computation of the consistency metric, we plan to explore alternative metrics or more efficient methods for assessing model consistency in our future work.

Additionally, to further enhance detection performance, future research could explore the integration of additional metrics, such as those related to learning dynamics (e.g., loss gradient) and prediction robustness, and develop efficient computational methods for their evaluation. Furthermore, subsequent studies could leverage the modeled hallucinations in the internal and output spaces to inform the development of additional strategies, such as data augmentation, feature selection, and training optimization, thereby advancing efforts to enhance the reliability of LLMs.

| D3T | MLP | NQ | | | | SciQ | | | |
|-----|-----|--------|--------|-------|-------|--------|--------|-------|-------|
| | | acc@50 ↑ | cov@50 ↑ | ECE ↓ | Brier ↓ | acc@50 ↑ | cov@90 ↑ | ECE ↓ | Brier ↓ |
| X-u | He | **0.355** | 0.120 | **0.026** | **0.119** | **0.766** | **0.228** | **0.076** | 0.200 |
| X-u | X-u | 0.351 | 0.117 | 0.030 | 0.123 | 0.763 | 0.224 | 0.079 | 0.202 |
| X-n | He | 0.354 | **0.121** | 0.028 | 0.120 | 0.765 | 0.226 | 0.077 | **0.198** |
| X-n | X-u | 0.350 | 0.118 | 0.031 | 0.123 | 0.761 | 0.223 | 0.080 | 0.201 |

Table 12: Performance variation with different initialization settings for the two hallucination detection networks.

## A.11 COMPLEXITY ANALYSIS

We analyze the size of the two hallucination detection networks utilized in the HADEMIF framework. The MLP model consists of two hidden layers, with the first hidden layer having a dimension of input_size × hidden_size, the second hidden layer having a dimension of hidden_size × hidden_size, and the final output layer having a dimension of hidden_size × vocab_size. Additionally, the trainable parameters of the D3T are primarily concentrated in the classifier, with the maximum dimension of the classifier weights being $(C + 1)^Q \times 2$, where $C$ denotes the maximum number of cut points and $Q$ represents the number of extracted prediction characteristics. Taking the Llama2-7B model as an example, training the MLP and D3T models requires only approximately 0.3% of additional parameters relative to Llama2-7B. In the case of GPT-2 XL (1.5B), only about 1.8% of the parameters relative to the GPT-2 XL are needed. This effectively demonstrates that the two hallucination detection networks are efficient in training and exhibit strong scalability.

## A.12 COMPUTATIONAL COSTS

We analyze the computational costs of our approach, which arise from the introduction of new modules and the computation of additional prediction characteristics. Taking Llama2-7B as an example, compared to fine-tuning LLMs using LoRA (with a rank of 8), the time required to update the two hallucination detection networks per iteration is only 0.9% of the time needed to update the LLMs. Furthermore, the time spent on characteristics extraction in each iteration is equivalent to the time required for updating the LLMs using LoRA. As observed, the majority of the time in our method is spent on characteristics extraction, particularly for the consistency metric. If the consistency metric is excluded, the time spent on characteristics extraction can be disregarded. Consequently, to enhance efficiency, we will consider utilizing alternative metrics or exploring more efficient methods for assessing model consistency in our future work.

## A.13 HALLUCINATION DETECTION CASES

We present examples of generated responses and their corresponding characteristic values using the GPT-2 XL model and the NQ dataset. For each question, three generated answers are provided. Five key distinguishing characteristics are reported, including maximum probability, Top-1 margin, Top-$K$ margin, uncertainty, and consistency. As shown in Table 13, non-hallucinated answers typically exhibit greater consistency, lower uncertainty, and larger margins. Moreover, the proposed D3T model effectively differentiates between hallucinated and non-hallucinated answers based on these extracted prediction characteristics.

**Question**: which country has the most coastline in the world
**GTAns**: Canada
**Ans1**: Italy
0.720966308, 0.544921872, 0.183104538, 0.186917218, 0.603609998
**Ans2**: Togo
0.496951332, 0.490139002, 0.149512008, 0.646159074, 0.669920596
**Ans3**: Canada
0.792032573, 0.809886966, 0.181390644, 0.149048981, 0.86631605
**Question**: who played doctor smith in lost in space
**GTAns**: Jonathan Harris
**Ans1**: Guy Williams
0.581436416, 0.483030135, 0.211672489, 0.522018769, 0.521877686
**Ans2**: Gary Richardson
0.775170857, 0.494472859, 0.188225367, 0.291994637, 0.668847207
**Ans3**: Jonathan Harris
0.986663492, 0.987808456, 0.212577534, 0.015423727, 0.732073851
**Question**: the joint between a coxal bone of the pelvis and the sacrum
**GTAns**: sacroiliac joint
**Ans1**: ischio iliacus
0.583019392, 0.577959439, 0.164734295, 0.568657438, 0.602541517
**Ans2**: Intertrochanteric
0.806365312, 0.822989212, 0.139759102, 0.199395772, 0.287201921
**Ans3**: sacroiliac joint
0.901740451, 0.910176217, 0.204612688, 0.065828692, 0.690894437
**Question**: what is a another name for the water cycle
**GTAns**: the hydrological cycle
**Ans1**: Great Circular Water Cycle
0.607422076, 0.641125625, 0.150578282, 0.521414703, 0.594664184
**Ans2**: hydrologic cycle
0.917772582, 0.92483196, 0.207370276, 0.055677697, 0.735232632
**Ans3**: Hydrologic Cycle
0.881650566, 0.891811086, 0.206473246, 0.079155331, 0.740394199
**Question**: dogs name in the grinch who stole christmas
**GTAns**: Max
**Ans1**: Maude
0.724961534, 0.508574101, 0.189241318, 0.380089306, 0.644346234
**Ans2**: Algie and Max
0.672750778, 0.700845736, 0.164279273, 0.505476371, 0.653290101
**Ans3**: Max
0.804949208, 0.813109496, 0.200413697, 0.102879738, 0.88730444
**Question**: what is the setting of the book hoot
**GTAns**: Florida
**Ans1**: Sumterville
0.883634505, 0.610795075, 0.19027741, 0.167938106, 0.630994929
**Ans2**: Pennsylvania, Singapore
0.507246027, 0.519549878, 0.153838168, 0.639813723, 0.87161937
**Ans3**: Florida
0.890308594, 0.898310994, 0.182605681, 0.081811693, 0.892968167

Table 13: Illustration of generated responses along with their corresponding characteristic values. The five values, presented from left to right, represent maximum probability, Top-1 margin, Top-$K$ margin, uncertainty, and consistency. All values are min-max normalized across the complete set for each characteristic. If multiple tokens are present in the generated answers, the mean of the metrics for those tokens is reported. Responses classified as hallucinations by the D3T model are indicated in red , whereas non-hallucinated responses are highlighted in green .

