# OpenReview forum: "HaDeMiF: Hallucination Detection and Mitigation in Large Language Models"
_ICLR.cc/2025/Conference — ICLR 2025 Poster_

### Official Review · Reviewer_goPV · 2024-10-17

**Soundness:** 2
**Presentation:** 3
**Contribution:** 2
**Rating:** 6
**Confidence:** 3

**Summary:**

The authors propose a framework for LLM hallucination detection and mitigation that consists of a deep neural decision tree (DNDT) that can detect hallucinations from the prediction characteristics of LLMs, and a multilayer perceptron (MLP) that can detect hallucinations from the internal states of LLMs. One can either calibrate the predictions of LLMs with the DNDT and MLP after optimizing them, or alternate between optimizing them and fine-tuning LLMs. In both cases, the DNDT and MLP are optimized to maximize the log-likelihood calibrated with them. The authors demonstrate the advantages of their framework by comparing its performance to those of existing model calibration methods.

**Strengths:**

- The authors benchmarked the proposed framework with a comprehensive suite of datasets.
- The paper is mostly easy to follow.

**Weaknesses:**

My primary concern is whether the proposed framework is indeed better than existing model calibration methods and plain fine-tuning of LLMs:
- The proposed framework introduces more tunable parameters than existing model calibration methods.
- The proposed framework is essentially a composition of three tunable models (a DNDT, an MLP, and an LLM), and I am unconvinced that tuning a composition of three tunable models can fundamentally better reduce hallucination than tuning one tunable model (an LLM). (Maybe some inductive biases in the DNDT help.) The empirical results can be strengthened by comparing the proposed framework against fine-tuning with varying numbers of parameters.

**Questions:**

I have raised my questions in Weaknesses.

---

> ### Author Response · Authors · 2024-11-20
> **Response to Reviewer goPV**
>
> Dear Reviewer goPV,
>
> We sincerely appreciate your thoughtful review and valuable feedback. We have carefully considered all of your comments and suggestions, and have made corresponding revisions to the manuscript. Below, we provide our detailed responses to each of your comments.
>
> > **Q1: The proposed framework introduces more tunable parameters than existing model calibration methods.**
>
> The volume of parameters introduced by our method is comparable to that of the LITCAB and P(IK) methods, while our approach consistently achieves superior performance. Moreover, while the label smoothing method requires fine-tuning the LLM, our method outperforms it even without the need for such fine-tuning.
>
> Following your valuable comments, to ensure a more thorough comparison with fine-tuning-based methods, we further compare HADEMIF with two other approaches: **Calibration-Tuning** [1] and **R-Tuning** [2], which fine-tunes the LLMs for model calibration and hallucination mitigation. The results of these comparisons are presented in **Tables 1** and **3** (**highlighted in blue**) of our revised manuscript. Although both Calibration-Tuning and R-Tuning fine-tune LLMs, their performance falls short of ours due to their inability to achieve fine-grained prediction calibration and their limitation to generating only binary confidence scores, respectively. All the above findings demonstrate the effectiveness of our approach in comparison to existing baselines.
>
> Additionally, we conducted a comparative analysis between the performance of our approach and the plain fine-tuning of LLMs using LoRA. The results for NQ and TruthfulQA, presented below, demonstrate that our approach consistently outperforms the plain fine-tuning method.
>
>
> |Dataset| NQ| | | |TruthfulQA| | | |
> |:--|:--:|:--:|:--:|:--:|:--:|:--:|:--:|:--:|
> |Method|acc@50|cov@50|ECE|Brier|acc@50|cov@50|ECE|Brier|
> |Original LLM|0.288|0.115|0.171|0.196|0.314 |0.136 |0.138 |0.218|
> |Label Smoothing|0.208|0.061|0.186|0.212|0.181 |0.000 |0.134 |0.175|
> |Temp. Scaling|	0.288|	0.115|0.165|	0.193|0.314| 	0.136| 	0.161| 	0.240 |
> |LITCAB|	0.300|	0.105|	0.101|0.169|	0.314 |0.195| 	0.105 | 	0.206|
> |LITCAB w/ Temp. Scaling|	0.300|0.105|0.083|	0.164|	0.314|	0.195|  	0.103 |	0.203|
> |Calibration-Tuning	|0.310|	0.115|	0.051|	0.142|	0.386|	0.393|	0.095|	0.198|
> |Plain Fine-tuning|	0.297|	0.110|	0.153|	0.186|	0.330|	0.201|	0.127|	0.200|
> |HADEMIF w/ Fine-tuning|	**0.355**|	**0.120**|	**0.026**|	**0.119**|**0.430**|**0.510**|**0.058**|**0.176**|
>
> Moreover, our experiments have demonstrated that simply increasing the number of tunable parameters does not always lead to performance improvement. As shown in **Figs. 3**(b) and (c) of the main text, optimal performance is achieved when the MLP network has 2 or 3 layers. However, increasing the number of layers to 4, while adding more tunable parameters, leads to performance degradation. Additionally, Fig. 3(d) indicates that increasing the maximum number of cut points in the D3T network does not necessarily enhance performance. These findings collectively suggest that the key to improving performance is not merely increasing the number of tunable parameters.
>
> > **Q2: The comparison of the performance against fine-tuning with varying numbers of parameters.**
>
> Thank you for your constructive suggestion. We have evaluated the performance of HADEMIF against fine-tuning with different numbers of parameters in LLMs. Specifically, we fine-tuned the Llama2-7B model using LoRA with rank values set to 8, 16, and 32. The corresponding results are presented in the table below.
>
> |Method|acc@50|cov@50|ECE|Brier|
> |:--|:--:|:--:|:--:|:--:|
> |Original   LLM|0.288|0.115|0.171|     0.196    |
> |Label   Smoothing|0.208|0.061|0.186|     0.212    |
> |Temp.   Scaling|0.288|0.115|0.165|     0.193    |
> |LITCAB|0.300|0.105|0.101|0.169|
> |LITCAB   w/ Temp. Scaling|0.300|     0.105    |     0.083    |     0.164    |
> |HADEMIF   (Rank 8)|     **0.355**  |     0.120    |     **0.026**    |     **0.119**    |
> |HADEMIF   (Rank 16)|     0.350  |     0.116    |     0.030    |     0.121    |
> |HADEMIF   (Rank 32)|     0.352    |     **0.125**    |     0.029    |     0.120    |
>
> The results show that HADEMIF consistently outperforms other approaches across different numbers of tunable parameters, underscoring its effectiveness under varying training complexities. We also observe that using a low rank in LoRA can already achieve strong performance, consistent with the findings presented in the original LoRA paper. These experiments have been added to Appendix A.5 in our revised manuscript.
>
> Moreover, from the results presented in Tables 6 and 7 of the manuscript, our method steadily demonstrates effectiveness across a wide range of model sizes (from 1.5B to **30B**) and architectures, highlighting its broad effectiveness.
>
> [1] Calibration-tuning: Teaching large language models to know what they don’t know.
>
> [2] R-tuning: Instructing large language models to say ‘I don’t know’.

---

> ### Author Response · Authors · 2024-11-23
> **Official Comment by Authors**
>
> Dear Reviewer goPV:
>
> We hope that our revisions and clarifications have adequately addressed your concerns. Your constructive feedback has been instrumental in enhancing the quality of our work. Should any aspects still require further elaboration, or if you have any additional questions, we would be pleased to address them. In light of the improvements and clarifications we have made, we would greatly appreciate it if you could consider adjusting your scores accordingly.
>
> Thank you once again for your time and effort in reviewing our work.
>
>
> Sincerely,
>
> Authors

---

> ### Author Response · Authors · 2024-11-26
> **Looking forward to your reply.**
>
> Dear Reviewer goPV:
>
> We sincerely appreciate the time and effort you have invested in reviewing our work. As we approach the close of the author-reviewer discussion period in one week, we kindly inquire whether our clarifications and revisions have sufficiently addressed your concerns. Your feedback on the current revision is extremely valuable, and we would be truly grateful for any further suggestions you may have to provide us with an additional opportunity to refine our work.
>
> Once again, we deeply appreciate your constructive comments and the contribution you have made to help strengthen our submission. Should any concerns remain, we would be more than happy to address them in full.
>
> Best regards,
>
> Authors

---

> ### Author Response · Authors · 2024-12-01
> **Gentle Reminder**
>
> Dear Reviewer goPV, we sincerely thank you for your thoughtful and thorough review of our manuscript, which has significantly strengthened the quality of our manuscript. As the discussion stage is ending soon, we would like to kindly inquire if you have any remaining questions or concerns. Any feedback is greatly appreciated and will undoubtedly help further enhance our work. Moreover, we would be truly grateful if you would consider increasing the score in light of the revisions and clarifications we have made, which were directly informed by your insightful feedback. Thanks again for your efforts in reviewing our paper!

---

> > ### Comment · Reviewer_goPV · 2024-12-02
> >
> > Thank you for addressing my concerns. I have revised my rating accordingly.

---

> > > ### Author Response · Authors · 2024-12-02
> > > **Official Comment by Authors**
> > >
> > > Dear Reviewer goPV:
> > >
> > > Thank you for raising the score! We are truly grateful for your recognition and encouraging feedback. Your valuable comments and constructive suggestions have greatly improved the quality of our manuscript. Once again, thank you for your invaluable contributions!
> > >
> > > Best regards,
> > >
> > > Authors

---

### Official Review · Reviewer_x16Z · 2024-10-23

**Soundness:** 3
**Presentation:** 2
**Contribution:** 2
**Rating:** 6
**Confidence:** 3

**Summary:**

The paper presents a new approach to addressing hallucinations in large language models, a significant issue where models generate nonfactual or incorrect information with high confidence. The proposed framework, HADEMIF, uses two neural networks: a Deep Dynamic Decision Tree (D3T) to detect hallucinations in the output space and a Multilayer Perceptron (MLP) to analyze hidden states in the internal semantic space. These models help calibrate the LLM’s predictions to reduce hallucinations.

**Strengths:**

1. A comprehensive approach that addresses both external outputs and internal token states for hallucination detection.
2. The framework introduces less than 2% additional parameters, making it lightweight.
3. Extensive experiments to show the reduced hallucination rates.

**Weaknesses:**

1. Shaky Methodology: The fundamental approach relies on using additional black-box models (D3T and MLP) to detect hallucinations in the LLM, which itself is a black box. This raises concerns about the interpretability and transparency of the method.

2. Unclear Training Data: The paper lacks clarity on where the training data for the D3T and MLP models come from. It is unclear whether hallucinations are labeled manually, generated synthetically, or detected through other processes. This makes it difficult to assess the generalizability of the framework.

3. Unfair Comparison with Training-Free Methods: The paper compares its method to training-free approaches, like self-consistency, without accounting for the difference in training requirements. A more balanced evaluation against similarly trained methods would provide a clearer picture of its effectiveness.

Overall, while the HADEMIF framework shows promise, these methodological concerns limit its potential impact and transparency.

**Questions:**

Here are a few questions and suggestions for the authors that could help clarify key points and potentially address limitations in the paper:

1. Training Data Source: Can you provide more detail on how the training data for the D3T and MLP models is collected? Specifically, how are hallucinated outputs labeled and validated? Is there a manual annotation process involved, or are automated methods used for labeling? A clearer explanation of the training data would improve understanding of the framework’s reliability and generalizability.

2. Interpretability of D3T: While the D3T model is described as interpretable, its role as a black-box model is still a concern. Could you provide additional insights or examples on how interpretability is achieved in practice? How does the model allow users to identify key characteristics that signal hallucinations?

3. Comparison with Training-Free Methods: Since you compare HADEMIF to training-free methods like self-consistency, could you clarify why these comparisons are appropriate given that HADEMIF requires additional training? How do you justify these comparisons, and would it be more fair to compare your method against other trained models?

4. Generalization Across Tasks: How well does HADEMIF generalize to different tasks and domains outside those tested in the experiments? Could you discuss whether the framework needs to be retrained or fine-tuned for new tasks, and if so, how this impacts the efficiency of the method?

5. Handling of Complex Scenarios: Can you provide more detail on how HADEMIF handles complex or ambiguous scenarios where there may be no clear factual answer (e.g., open-ended creative writing tasks)? Is the framework likely to flag creative responses as hallucinations in these cases?

---

> ### Author Response · Authors · 2024-11-20
> **Response to Reviewer x16Z (Part I)**
>
> Dear Reviewer x16Z,
>
> We sincerely appreciate your insightful review and valuable comments and suggestions. We have carefully considered each of them and made corresponding revisions in the updated version of our manuscript. Below, we provide our detailed responses to your comments:
>
> > **Q1: Training data source.**
>
> We apologize for not clearly explaining the training data of the two hallucination detection networks in the main text, which may cause confusion for the readers. Our approach does not require additional annotations for hallucinations, meaning that no manual or automatic labeling process is involved. The two detection networks are optimized using the original training data from the dataset described in Appendix A.1 by minimizing the loss of the LLMs, with predictions calculated using Eq. (3) in the main text. During training, these two networks capture hallucination biases, such as overconfidence and inaccuracies, in both the output and internal spaces of the LLMs and generate adjustment terms to calibrate the LLMs’ prediction distribution, with the goal of maximizing the token probabilities for correct generations while minimizing the likelihood of incorrect ones.
>
> We have clarified this point in **Section 3.1.2** of the revised version of our manuscript (**highlighted in blue**). The added content is as follows: "*Notably, our method does not require any additional hallucination annotations. It only requires training the two hallucination detection networks by minimizing the loss of LLMs on the original training set, with predictions computed as outlined in Eq. 3. This process aims to maximize the token probabilities for correct generations while reducing the likelihood of incorrect ones.*"
>
> > **Q2: Interpretability of D3T.**
>
> We humbly respond to this comment from the following aspects:
> - **How D3T achieves its interpretability:** The interpretability of D3T arises from its tree-based structure, which employs a soft binning function for feature splitting and utilizes the Kronecker product operation to determine the final node of the tree. This tree structure enables the revelation of decision rules for hallucination detection, which reflect how the combination of prediction characteristics influences hallucination detection. Although the importance of various input features is not explicitly incorporated during the training process of D3T, the method’s reliance on feature splitting enables the computation of their information gains, which can be used to assess their significance, thereby highlighting key characteristics for hallucination detection.
> - **Conducted analysis for its interpretability:** In Appendix A.7, we visualized the first three layers of the decision tree in Fig. 5, where each path represents a hallucination detection rule, illustrating how the combination of various prediction characteristics determines the existence of hallucination. The findings reveal that hallucinated generations are generally associated with low consistency, high uncertainty, and small margins. Additionally, in Section 4.6, we assess the importance of various prediction characteristics (Fig. 3(a)) by calculating their information gains. The results indicate that consistency and uncertainty are the most significant factors, while the additional characteristics we introduced also contribute meaningfully to the detection process.
> - **Broader impact for future research:** We believe that a thorough exploration of the derived decision rules could inform future research directions, such as refining directions for improving LLMs, enhancing user trust and acceptance, and enabling greater control over LLMs through real-time monitoring and analysis. These will be the focus of our future research.
>
> Based on your valuable feedback, we have included additional explanations about the interpretability of D3T in Section 3.1.1, Section 4.6, and Appendix A.7 in our revised manuscript.

---

> ### Author Response · Authors · 2024-11-20
> **Response to Reviewer x16Z (Part II)**
>
> > **Q3: Comparison with more training-based approaches.**
>
> We completely agree that incorporating additional training-based approaches would enhance the fairness of our comparison. Our previous experiments have already included several such training-based methods. Specifically, the label smoothing method involves fine-tuning the LLMs using LoRA with the label smoothing loss. Additionally, the P(IK) and LITCAB methods necessitate the training of additional linear layers that are introduced. The rationale for comparing both training-based and non-training-based methods lies in previous studies, which suggest that, in the context of model calibration and hallucination, a training-based method is not always superior to a non-training method. Thus, we believe that incorporating comparisons across different types of methods is meaningful.
>
> For a more thorough comparison, we further incorporated two other fine-tuning methods for LLMs: **R-Tuning** [1] and **Calibration-Tuning** [2], which fine-tunes the LLMs for hallucination mitigation and calibration, respectively. Since Calibration-Tuning produces a single score for the entire generation, it cannot evaluate calibration at the claim level, as each generation typically includes multiple claims. Therefore, we do not assess its performance on paragraph-level tasks, following LITCAB. The results of these additional comparisons are presented in **Tables 1** and **3** (highlighted in blue) of our revised manuscript. Although both Calibration-Tuning and R-Tuning fine-tune LLMs, their performance falls short of ours due to their inability to achieve fine-grained prediction calibration and their limitation to generating only binary confidence scores, respectively.
>
> > **Q4: Generalization across tasks.**
>
> As the two hallucination detection networks exhibit a certain degree of generalization across various tasks, applying our approach to a new task requires only fine-tuning the networks, rather than retraining them from scratch. To illustrate this, we conduct experiments on the NQ and TriviaQA datasets, using detection networks pre-trained on the SciQ dataset. We evaluate performance under three conditions: no fine-tuning, fine-tuning the two hallucination detection networks for 5 epochs, and fine-tuning for 15 epochs. As shown in the table below, only fine-tuning for 5 epochs can achieve reasonable performance, while fine-tuning for 15 epochs results in performance nearly identical to that obtained from training the model from scratch.
>
> |Dataset|NQ| | | |TrivalQA| | | |
> |:--|---------------|:--:|:--:|:--:|:--:|:--:|:--:|:--:|
> |Metric|acc@50|     cov@50    |     ECE      |     Brier    |     acc@50      |     cov@60    |     ECE      |     Brier    |
> |From   scratch|  **0.315**|**0.115**|**0.034**|**0.116**|     0.480       |**0.234**|**0.080**|**0.190**|
> |5   epoch|0.307|0.111|     0.049    |0.130    |     0.476       |     0.221     |     0.083    |     0.199    |
> |15   epoch|0.314|**0.115**|0.036|0.119| **0.483**|**0.234**|**0.080**|     0.192    |
>
> The need for additional training time is a common drawback of methods that introduce extra modules, such as adapters in the parameter-efficient fine-tuning of LLMs and the LITCAB method. However, the generalization ability of our approach across various tasks partially mitigates this limitation. Moreover, the generalization ability of the two detection networks highlights the potential for developing a universal hallucination detection network by training on large, diverse datasets. We intend to investigate this avenue in our future research.
>
> > **Q5: Handling complex scenarios, such as open-ended creative writing tasks, where no clear factual answer exists.**
>
> Our approach introduces two hallucination detection networks to capture hallucination biases, such as overconfidence and inaccuracy, in both the output and internal spaces of LLMs. These networks then generate logit adjustment terms to calibrate the LLMs’ prediction distribution, thereby eliminating hallucinations and ensuring more reliable predictions.
>
> In scenarios where true answers are unavailable, such as in open-ended creative writing tasks, self-supervised training objectives should be utilized, with the predicted probability distribution being calibrated using the outputs from the two hallucination detection networks, like Eq. (3) of the main text. As you mentioned, creative content may exhibit relatively high uncertainty and low consistency, which could lead to it being identified as hallucinations. To fully mitigate this issue, it may be necessary to incorporate additional strategies, such as human feedback or the integration of more creative data, during the training of the hallucination detection networks. We will explore the effectiveness of our approach on open-ended tasks in future research.
>
> [1] R-tuning: Instructing large language models to say ‘I don’t know’.
>
> [2] Calibration-tuning: Teaching large language models to know what they don’t know.

---

> > ### Comment · Reviewer_x16Z · 2024-11-21
> >
> > Thank you for your responses and the updates to your manuscript. I appreciate the clarifications and additional experiments you've provided. Here are a few more comments and suggestions to strengthen your work:
> >
> > 1. Preference Learning on Datasets
> > Consider using preference learning methods, such as DPO, directly on your datasets to train the LLMs. This could help the model better avoid hallucinations by aligning its outputs with human judgments. Preference learning might capture subtle aspects of hallucination detection that calibration networks alone may miss.
> >
> > 2. Interpretability of the Decision Tree (D3T)
> > Clarification Needed: The interpretability of D3T relies on understanding which features correspond to each tree node. Without this, it's hard to interpret the model. Include diagrams or tables showing which features are used at each node and how they affect decisions. This will make your model's reasoning clearer.

---

> > > ### Author Response · Authors · 2024-11-22
> > > **Response to Reviewer x16Z**
> > >
> > > Dear Reviewer x16Z,
> > >
> > > Thank you for your prompt reply. We sincerely appreciate your thoughtful comments and valuable suggestions. Below, we provide our detailed responses to your new comments.
> > >
> > > > **1. Preference learning utilization, such as DPO.**
> > >
> > > Thank you for your insightful comments. Our approach and preference learning algorithms, such as DPO, can be considered as two complementary avenues for improving the generation quality of LLMs. DPO optimizes LLMs by minimizing the DPO loss on an offline constructed preference dataset, thereby aligning the generated responses more closely with human preferences. In contrast, our approach models the hallucination bias inherent in the generation process through two networks and uses the modeled bias to adjust the predictions of LLMs, thereby mitigating hallucinations.
> > >
> > > Both methods can be seen as improvements to the optimization objective of LLMs. However, compared to DPO, our approach circumvents the costly and time-consuming steps of constructing offline preference datasets and training reference models. Moreover, our method offers a more direct and effective approach for hallucination detection and mitigation compared to DPO, as judging hallucinations involves not only 'preferences' but also requires that the model's output aligns with ground-truth answers or real-world knowledge.
> > >
> > > Nevertheless, we strongly agree with you that an organic integration of these two methods could lead to even greater improvements in LLM generation quality and plan to explore this integration in our future research.
> > >
> > > > **2. Interpretability of D3T.**
> > >
> > > D3T can be viewed as a model composed of a series of decision rules, where it begins by segmenting each input feature using soft binning and then combines features that satisfy various conditions using the Kronecker product. Therefore, it does not have an explicit hierarchical structure in the computation process and relies solely on a series of decision rules. Following your valuable feedback, we have included a detailed list of fifty decision rules learned by D3T in **Appendix A.7** of the revised manuscript (**highlighted in orange**), as shown in **Table 11**. These rules illustrate how various combinations of characteristics contribute to the detection of hallucinations. Our analysis indicates that low consistency, high uncertainty, small margins, and low maximum probability are typically indicative of hallucinations, which aligns with prevailing perceptions. However, the decision rules also reveal that this relationship is not absolute. As such, relying solely on a single metric, as has been done in prior methods, is insufficient for effectively detecting hallucinations.
> > >
> > > Moreover, to provide a more intuitive presentation of the model's decision-making process and better account for the contribution of each feature, we introduced the concept of information gain to assess the importance of individual characteristics and visualized the decision tree, where the hierarchical structure of the characteristics is arranged in descending order of their information gains. Appendix **Fig. 5** presents the resulting decision tree, which illustrates the features at each node and demonstrates how they contribute to the decision rules for hallucination detection.
> > >
> > > Thanks again for your thoughtful responses and comments. We hope that our responses and revisions have adequately addressed your concerns. If you have any further questions or suggestions, please feel free to share them.

---

> > > > ### Comment · Reviewer_x16Z · 2024-11-25
> > > >
> > > > Thanks for addressing the concerns. I have revised the scores accordingly.

---

> > > > > ### Author Response · Authors · 2024-11-26
> > > > > **Official Comment by Authors**
> > > > >
> > > > > Dear Reviewer x16Z:
> > > > >
> > > > > Thank you for increasing the score! Your valuable suggestions have significantly contributed to improving the quality of our manuscript. Again, we sincerely thank you for your precious time and thoughtful comments! We would also greatly appreciate any additional feedback or suggestions that could further enhance the manuscript and potentially lead to further score improvement.
> > > > >
> > > > > Sincerely,
> > > > >
> > > > > Authors

---

> ### Author Response · Authors · 2024-11-23
> **Official Comment by Authors**
>
> Dear Reviewer x16Z:
>
> We hope that our clarifications and revisions have adequately addressed your concerns. We believe the paper has been significantly improved thanks to your comments. If any points remain unclear, or if you have additional questions or suggestions, please feel free to share them. Given the improvements and clarifications we have made, we would be most grateful if you could consider adjusting your scores accordingly.
>
> Thank you once again for your time and effort in reviewing our work.
>
> Sincerely,
>
> Authors

---

### Official Review · Reviewer_QEQq · 2024-10-30

**Soundness:** 2
**Presentation:** 3
**Contribution:** 2
**Rating:** 6
**Confidence:** 3

**Summary:**

This study introduces HADEMIF, a comprehensive framework for hallucination detection and mitigation in large language models (LLMs). HADEMIF leverages both the output space and internal hidden states of LLMs through two compact networks to effectively identify and reduce hallucinations. The framework is applicable in both inference and fine-tuning stages, and experiments demonstrate its superior performance in hallucination detection and calibration across various open-source LLMs compared to existing methods.

**Strengths:**

1. The authors integrate multiple metrics to simultaneously assess hallucination in both output space and hidden states, achieving a certain degree of interpretability.
2. The author considers hallucination at multiple levels in the experiments and employs various LLM backbones.

**Weaknesses:**

1. Is there a direct relationship between calibration and hallucination? This gap has not been clearly addressed. Although some related work is cited in line 45, this key issue underpinning the entire paper has not been convincingly articulated. Furthermore, we observe in the experimental results that the ECE metric, representing calibration, and the acc@50 metric, representing hallucination, show an inverse relationship to some extent in certain methods.

2. In the authors' statement on line 165, they suggest that single metrics may limit the effectiveness of hallucination detection, but this is not substantiated in the paper. To our knowledge, the authors' method can be summarized as manually extracting certain features and feeding them into a deep decision tree, with these features having appeared to some extent in previous work. Therefore, we have reason to doubt that this combination effectively addresses the challenge of discovering metrics for hallucination detection.

3. The authors use an improved DNDT for hallucination detection in the paper, but they did not conduct related ablation experiments to assess the performance difference between D3T and DNDT.

4. During the training process, it appears that the LLM evolves in conjunction with the detector. However, the authors did not specify the initialization of the detector's weights, which we believe could lead to incorrect gradients being applied to the LLM initially, thereby reducing its generalizability. We also note that the authors did not conduct related experiments on this matter.

5. The authors emphasized interpretability. However, they only provided the impact of various metrics on the output space in the experiments, without focusing on the interpretability of metric itself.

6. The authors seemingly did not compare their method with finetune-based methods or methods specifically designed for hallucination mitigation in their experiments. Instead, they only compared it with some calibration methods and lacked data on finetuning the backbone on these training sets. This omission may lead to unfair comparisons when introducing training in their method, and the comparison methods might also be relatively weak.

7. It would be beneficial to provide experimental results on the computational costs of the model due to the introduction of new modules and the calculation of new metrics, especially consistency.

**Questions:**

Refer to weakness.

---

> ### Author Response · Authors · 2024-11-20
> **Response to Reviewer QEQq (Part I)**
>
> Dear Reviewer QEQq,
>
> We deeply appreciate your thorough review and valuable comments. We have carefully considered all your comments and have addressed them in the revised manuscript. Below are our detailed responses to each of your comments.
>
> > **Q1: The gap between calibration and hallucination and the relationship between ECE and acc@50.**
>
> We respectfully address this concern from the following perspectives:
>
> - **The relationship between hallucination and calibration:** Their relationship is mutually reinforcing. Calibrating LLMs plays a crucial role in detecting and mitigating hallucinations, while the mitigation of hallucinations, in turn, contributes to enhancing model calibration. In our approach, two lightweight networks are introduced to capture hallucinations in both the output and internal spaces of LLMs and generate adjustment terms to calibrate the model’s probability distribution, with the goal of maximizing the token probabilities for correct generations while minimizing the likelihood of incorrect ones, thereby achieving both hallucination mitigation and model calibration.
> - **Further clarification in our manuscript:** We have incorporated the aforementioned content into Section 1 of our revised manuscript. Moreover, to further clarify the relationship between hallucination mitigation and model calibration, we have also expanded the related work section (Section 2) to include additional discussions on model calibration (**highlighted in blue**).
> - **Relationship between ECE and acc@50:** Based on the performance improvements observed across all four metrics, our approach demonstrates effectiveness in both hallucination mitigation and model calibration. The two metrics you mentioned, ECE and acc@50, evaluate model performance from different perspectives. There is no strict proportional or inverse relationship between them, as indicated by their calculation manners.
> - **Additional experiments for hallucination detection:** To provide a more comprehensive evaluation of the hallucination detection capability of our approach, we compared it against three additional detection baselines using the LLaMA-7B model. For our approach, the hallucination detection score is derived from the D3T model. The evaluation metrics include two widely adopted hallucination detection indicators: AUROC and PCC. As shown in the results below, our method consistently outperforms previous detection approaches that rely solely on a single characteristic. These experiments have been included in Appendix A.4 of our revised manuscript.
>
> |     Dataset                     |     NQ              |                     |                 |     TrivalQA        |                  |              |
> |:--|:--:|:--:|:--:|:--:|:--:|:--:|
> |     Metric                      |     AUC$_{s}$       |     AUC$_{r}$       |     PCC         |     AUC$_{s}$       |     AUC$_{r}$    |     PCC      |
> |     Perplexity   [1]            |     0.740           |     0.747           |     0.301       |     0.836           |     0.836        |     0.544    |
> |     LN-Entropy   [2]            |     0.728           |     0.737           |     0.298       |     0.834           |     0.832        |     0.540    |
> |     Lexical   Similarity [3]    |     0.738           |     0.759           |     0.306       |     0.826           |     0.840        |     0.556    |
> |     HADEMIF                     |     **0.772**           |     **0.776**           |     **0.389**       |     **0.843**           |     **0.842**        |     **0.581**    |
>
>
> [1] Out-of-distribution detection and selective generation for conditional language models.
>
> [2] Uncertainty estimation in autoregressive structured prediction.
>
> [3] Towards collaborative neural-symbolic graph semantic parsing via uncertainty.

---

> ### Author Response · Authors · 2024-11-20
> **Response to Reviewer QEQq (Part II)**
>
> > **Q2: The effectiveness of the combination of a series of prediction characteristics.**
>
> We humbly respond to this comment from the following aspects:
>
> - **The limitation of utilizing single metrics and the superiority of using a combination of characteristics:** In the response to Question 1, our method consistently achieves higher AUROC and PCC values compared to those based on individual metrics, highlighting that reliance on a single characteristic is insufficient for achieving satisfactory detection performance. Moreover, the detection cases presented in Table 12 further underscore the limitations of using a single metric for hallucination detection. For instance, prior studies have often assumed that predictions with high uncertainty and low consistency are indicative of hallucinations. However, as shown in Table 12, even generations exhibiting high consistency and low uncertainty can still manifest hallucinations. Additionally, the results in Table 3 demonstrate that our approach consistently outperforms previous hallucination detection and mitigation baselines, validating the effectiveness of leveraging a combination of characteristics.
> - **The rationale behind characteristics selection:** Although some of our adopted prediction characteristics have appeared in previous studies, to the best of our knowledge, no hallucination detection method has yet integrated these characteristics and used a network to automatically learn the function between these metrics and hallucinations. Based on our findings, this approach achieves superior performance and provides new insights for future hallucination research. To further clarify the rationale behind our selection of these characteristics, we have provided additional explanations in Section 3.1.1 of our revised manuscript. The revised content is as follows: "*First, we consider the commonly used hallucination detection metrics from previous studies, specifically uncertainty and consistency. Next, we incorporate three additional metrics that reflect prediction confidence and are commonly employed in previous machine learning tasks, such as fairness evaluation and sample weighting (Zhang et al., 2020; Ross & Doll’ar, 2017; Jin et al., 2024).*"
> - **Significance of our extracted characteristics:** In Section 4.6, we have evaluated the significance of our extracted characteristics by calculating their respective information gains, as illustrated in Fig. 3(a). The results demonstrate that all metrics contribute to the hallucination detection process, with consistency and uncertainty proving to be particularly influential.
>
> > **Q3: Ablation studies for the comparison between D3T and DNDT.**
>
> We fully agree with your suggestion to conduct an ablation study comparing D3T with DNDT. Unlike DNDT, which employs a fixed architecture resulting in numerous redundant branches, our D3T network features a flexible structure that dynamically adjusts the number of cut bins for each input feature during training. This adaptability allows for better alignment with the local characteristics of the data, thereby reducing the risk of overfitting. The effectiveness of both models as hallucination detection networks has been compared on the NQ and SciQ datasets. The results indicate that employing D3T as the hallucination detection network consistently outperforms the DNDT model. These experiments have been included in Appendix A.6 of the revised version of our manuscript.
>
> |     Datasets    |     Metric    |     HADEMIF   with D3T    |     HADEMIF   with DNDT    |
> |:-----------------|:---------------|:---------------------------:|:----------------------------:|
> |     NQ          |     acc@50    |     **0.315**                 |     0.311                  |
> |                 |     cov@50    |     **0.115**                 |     0.112                  |
> |                 |     ECE       |     **0.034**                 |     0.045                  |
> |                 |     Brier     |     **0.116**                 |     0.131                  |
> |     SciQ        |     acc@50    |     **0.760**                 |     0.754                  |
> |                 |     cov@90    |     **0.230**                 |     0.225                  |
> |                 |     ECE       |     **0.083**                 |     0.084                  |
> |                 |     Brier     |     **0.201**                 |     0.203                  |

---

> ### Author Response · Authors · 2024-11-20
> **Response to Reviewer QEQq (Part III)**
>
> > **Q4: Initialization of the weights of the two hallucination detection networks.**
>
> We apologize for the omission of the initialization settings for the two hallucination detection networks in the original manuscript. Corresponding contents have been added to Appendix A.2 of our revised manuscript (highlighted in blue). For the MLP network, we utilize He initialization [1], as this initialization approach is well-suited for layers with ReLU activation functions. For the D3T model, following the initialization approach in DNDT, all parameters are initialized using Xavier initialization with a uniform distribution [2].
>
> To identify the optimal initialization settings for the two networks, we compared four different initialization configurations under the condition of the same number of training iterations (40 epochs). The MLP network considers two initialization methods: He initialization and Xavier initialization with a uniform distribution (denoted as X-u). Moreover, D3T considers two initialization methods: X-u and Xavier initialization with a normal distribution (denoted as X-n). The comparison results are shown in the table below:
>
> |     D3T    |     MLP    |     NQ        |               |              |              |     SciQ      |               |              |              |
> |:------------:|:------------:|:---------------:|:---------------:|:--------------:|:--------------:|:---------------:|:---------------:|:--------------:|:--------------:|
> |            |            |     acc@50    |     cov@50    |     ECE      |     Brier    |     acc@50    |     cov@90    |     ECE      |     Brier    |
> |     X-u    |     He     |     **0.355**     |     0.120     |     **0.026**    |     **0.119**    |     **0.766**     |     **0.228**     |     **0.076**    |     0.200    |
> |     X-u    |     X-u    |     0.351     |     0.117     |     0.030    |     0.123    |     0.763     |     0.224     |     0.079    |     0.202    |
> |     X-n    |     He     |     0.354     |     **0.121**     |     0.028    |     0.120    |     0.765     |     0.226     |     0.077    |     **0.198**    |
> |     X-n    |     X-u    |     0.350     |     0.118     |     0.031    |     0.123    |     0.761     |     0.223     |     0.080    |     0.201    |
>
> Based on the above results, He initialization, specifically designed for ReLU activation functions, proves to be more advantageous for initializing the MLP network. For the D3T network, the performance difference between Xavier initialization with uniform and normal distributions is minimal. The above experiments have been added to Appendix A.9 of our revised manuscript.
>
> > **Q5: The interpretability of metrics.**
>
> The interpretability of our approach arises from the use of an interpretable model (i.e., D3T) for the output-space hallucination detection network, which facilitates the revelation of the decision rules for hallucination detection associated with the prediction characteristics (as shown in Fig. 5 of the Appendix). Furthermore, the relative importance of various characteristics in hallucination detection can be analyzed by calculating their information gains thanks to its feature-splitting operation (as shown in Fig. 3(a) in the main text).
>
> Based on our understanding, the interpretability of the metrics you referenced appears to be related to how the state of each characteristic correlates with the occurrence of hallucinations. According to our analysis, there is no definitive conclusion on this matter. For instance, previous studies that rely solely on a single metric often assume that high consistency or low uncertainty implies the absence of hallucinations. Yet, as demonstrated in Table 12 of our manuscript, this assumption does not always hold, as generations with high consistency or low uncertainty can still be hallucinations. Nevertheless, the decision rules obtained from the D3T model can reflect how the combination of various prediction characteristics determines the presence of hallucinations. Therefore, extensive analysis of these rules can uncover certain patterns. From our analytical experiments, non-hallucinated answers typically exhibit high consistency, low uncertainty, and large margins, although this relationship is not absolute. We are open to further clarification if our explanation differs from your intention.
>
> [1] Delving deep into rectifiers: Surpassing human-level performance on ImageNet classification.
>
> [2] Understanding the difficulty of training deep feedforward neural networks.

---

> ### Author Response · Authors · 2024-11-20
> **Response to Reviewer QEQq (Part IV)**
>
> > **Q6: Comparison with finetune-based methods and methods designed for hallucination mitigation.**
>
> Our previous comparison has included some training-based methods. For example, the label smoothing approach that we have compared involves fine-tuning the LLMs using LoRA with the label smoothing loss (as detailed in lines 376-377 of the main text). Moreover, the P(IK) and LITCAB methods necessitate the training of additional linear layers that are introduced.
>
> Following your valuable feedback, to provide a more comprehensive and balanced comparison, we further compared our approach with two other fine-tuning methods for LLMs: **R-Tuning** [1], which fine-tunes the model to mitigate hallucinations, and **Calibration-Tuning** [2], which fine-tunes the model to improve calibration. Since Calibration-Tuning produces only a single unified score for the entire generation, it cannot evaluate calibration at the claim level, as each generation typically contains multiple claims. Consequently, following LITCAB, we do not evaluate the performance of Calibration-Tuning for paragraph-level tasks. The results of these additional comparisons are presented in **Tables 1** and **3** (highlighted in blue) of our revised manuscript. Although both Calibration-Tuning and R-Tuning fine-tune LLMs, their performance falls short of ours due to their inability to achieve fine-grained prediction calibration and their limitation to generating only binary confidence scores, respectively.
>
> > **Q7: Computational costs.**
>
> Taking Llama2-7B as an example, compared to fine-tuning LLMs using LoRA (with a rank of 8), the time required to update the two hallucination detection networks per iteration is only **0.9%** of the time needed to update the LLMs. Moreover, the time required for characteristics extraction per iteration is equivalent to the time needed to update the LLMs using LoRA. As observed, the majority of the time in our approach is spent on characteristics extraction, particularly for the calculation of the consistency metric. If the consistency metric is excluded, the time spent on feature extraction per iteration reduces to less than 0.1% of the time required for updating the LLMs using LoRA. Consequently, to improve efficiency, we plan to explore alternative metrics and more efficient methods for assessing model consistency in future work. We have added the above details regarding the computational overhead of our approach in Appendix A.12 of the revised manuscript.
>
> [1] R-tuning: Instructing large language models to say ‘I don’t know’.
>
> [2] Calibration-tuning: Teaching large language models to know what they don’t know.

---

> ### Author Response · Authors · 2024-11-23
> **Official Comment by Authors**
>
> Dear Reviewer QEQq:
>
> We hope that our explanations and revisions have adequately addressed your concerns. All of your comments are valuable and have been taken into careful consideration and fully addressed in the revisions. If you find them satisfactory, we kindly ask that you take these improvements into account when reconsidering the score. We remain open to any further comments, follow-up questions, or suggestions, and would greatly appreciate your feedback.
>
> Thank you once again for your time and effort in reviewing our work.
>
> Sincerely,
>
> Authors

---

> ### Author Response · Authors · 2024-11-26
> **Looking forward to your reply.**
>
> Dear Reviewer QEQq:
>
> We sincerely appreciate the time and effort you have invested in reviewing our work. As we approach the close of the author-reviewer discussion period in one week, we kindly ask whether you are satisfied with the clarifications and revisions we have made. Your feedback on the current revision is extremely valuable, and we would greatly appreciate any further suggestions you may have to provide us with an additional opportunity to refine our work.
>
> Once again, we would like to express our sincere gratitude for your thoughtful review and the contributions you have made to strengthening our submission. Should any further concerns remain, we are fully committed to addressing them.
>
> Best regards,
>
> Authors

---

> ### Comment · Reviewer_QEQq · 2024-11-27
> **Raising the Score.**
>
> i’d like to thank the authors for the response. Your experiments have clarified most of my doubts. I have modified the score.

---

> > ### Author Response · Authors · 2024-11-27
> > **Thank you for your acknowledgment.**
> >
> > Dear Reviewer QEQq:
> >
> > Thank you for raising the score! We sincerely appreciate your acknowledgment and encouraging feedback. Your valuable comments and constructive suggestions have significantly contributed to improving the quality of our paper. Thank you once again for your invaluable contributions!
> >
> > Best regards,
> >
> > Authors

---

### Official Review · Reviewer_76SS · 2024-11-01

**Soundness:** 3
**Presentation:** 3
**Contribution:** 3
**Rating:** 6
**Confidence:** 5

**Summary:**

This paper introduces HADEMIF, a framework designed to detect and mitigate hallucinations in large language models (LLMs). It integrates a novel interpretable Deep Dynamic Decision Tree (D3T) for output space analysis and a Multilayer Perceptron (MLP) for internal state analysis. HADEMIF effectively calibrates LLM predictions with minimal parameter increase, enhancing model reliability.

**Strengths:**

- Originality: Novel use of D3T and MLP for multifaceted hallucination detection.
- Quality: Demonstrated robustness through extensive evaluations, including accuracy and Expected Calibration Error (ECE).
- Clarity: Mostly lear methodology, though technical details may be challenging for broader audiences.

**Weaknesses:**

- Some parts in the method lack explanations. For example, before line 184 "Following the approach of DNDT, D3T replaces ..." there should be clearer explanations for the DNDT method
- Limited explanation of the interpretability benefits of D3T in practical application.

**Questions:**

- Could the authors clarify the choice of metrics (such as consistency [1]) and their weights within D3T? (Line 209 "Extraction of Prediction Characteristics"). Better motivated metric selection would facilitate the understanding of the numeric values for these metrics.
- Is there a threshold for model applicability when training complexity increases, particularly with LoRA?

[1] Better to Ask in English: Cross-Lingual Evaluation of Large Language Models for Healthcare Queries.

---

> ### Author Response · Authors · 2024-11-20
> **Response to Reviewer 76SS (Part I)**
>
> Dear Reviewer 76SS,
>
> We express our sincere gratitude for your thorough review and insightful comments on our paper. We have carefully considered all your comments and addressed them in our revised manuscript. Below are our specific replies to your comments:
>
> > **Q1: More explanations about DNDT.**
>
> We fully acknowledge your suggestion to provide a more detailed explanation of the DNDT model. In response, we expanded the description of DNDT in **Section 3.1.1** of our revised manuscript (**highlighted in blue**). The revised contents are as follows: "*D3T is an adaptation of the Deep Neural Decision Tree (DNDT) model (Yang et al., 2018), which is a tree model implemented using a neural network. This method introduces a soft binning function for feature splitting, which is achieved through a linear layer with Softmax as the activation function, and utilizes the Kronecker product operation to determine the final node of the tree.*"
>
> Moreover, to systematically compare the performance of DNDT and D3T, we conducted ablation studies using both networks as hallucination detection networks for the output space. The experimental results, presented below, demonstrate that employing the D3T model as the hallucination detection network yields superior performance, owing to its flexible structure during training.
>
> |Dataset|     Metric    |     D3T    |     DNDT    |
> |:-----|:-----|:-----:|:-----:|
> |NQ|     acc@50    |     **0.315**                 |     0.311                  |
> | |     cov@50    |     **0.115**                 |     0.112                  |
> | |     ECE       |     **0.034**                 |     0.045                  |
> | |     Brier     |     **0.116**                 |     0.131                  |
> |     SciQ       |     acc@50    |     **0.760**                 |     0.754                  |
> | |     cov@90    |     **0.230**                 |     0.225                  |
> | | ECE |     **0.083**                 |     0.084                  |
> | | Brier |     **0.201**                 |     0.203                  |
>
> These experiments have been included in Appendix A.6 of our revised manuscript. Additionally, we have carefully reviewed and revised the entire manuscript to ensure that all descriptions are clear and accurate.
>
> > **Q2: Limited explanation of the interpretability benefits of D3T in practical application.**
>
> We humbly respond to this comment from the following aspects:
>
> - **Challenges that can be solved:** Existing studies employ different predictive characteristics for hallucination detection, yet the relative importance of various metrics remains unclear. Moreover, these methods often rely on manually defined functions or thresholds, lacking proper guidance.
> - **How D3T achieves its interpretability:** The interpretability of D3T arises from its tree-based structure, which employs a soft binning function for feature splitting and utilizes the Kronecker product operation to determine the final node of the tree. This tree structure enables the revelation of decision rules for hallucination detection, which reflect how the combination of prediction characteristics influences hallucination detection. Although the importance of various input features is not explicitly incorporated during the training process of D3T, due to its reliance on feature splitting, the information gain for various characteristics can be computed to reflect their importance. Consequently, when visualizing the tree, the characteristics can be organized in layers according to their importance, in descending order.
> - **Conducted analyses using its interpretability:** In Appendix A.7, we visualized the first three layers of the decision tree (Fig. 5), where each path represents a decision rule reflecting how the combination of prediction characteristics determines the existence of hallucinations. From the results, hallucinated generations are generally associated with low consistency, high uncertainty, and small margins. Additionally, in Section 4.6, we visualize the importance of various characteristics in Fig. 3(a). The results indicate that while uncertainty and consistency play a dominant role, other predictive characteristics, such as margin and predicted probability, are also useful for hallucination detection.
> - **Benefits and broader impact:** The interpretability of D3T can facilitate a clearer understanding of the relationship between predictive characteristics and hallucination detection, as well as an analysis of which metrics are most critical for detecting hallucinations. We believe that a thorough exploration of the derived decision rules could offer valuable guidance for future research, such as refining the directions for improving LLMs and establishing stronger control over LLMs through real-time monitoring and analysis.
>
> Building on the responses above, we have provided further explanations regarding the interpretability of D3T in Section 3.1.1, Section 4.6, and Appendix A.7 of the revised manuscript.

---

> ### Author Response · Authors · 2024-11-20
> **Response to Reviewer 76SS (Part II)**
>
> > **Q3: Clarification of the choice of metrics and their weights within D3T.**
>
> Previous studies typically utilize uncertainty and consistency for hallucination detection, generally assuming that hallucinations are associated with high uncertainty and low consistency. Our approach continues to leverage these two metrics without any assumptions. Additionally, to capture knowledge within the output space more comprehensively, we explore three other indicators: predicted probability distribution, predicted margin, and logits vector. Generally, a large maximum predicted probability, a wide margin, and a large norm of the logits vector indicate high confidence in the prediction. These metrics that reflect prediction confidence are commonly applied in previous machine learning tasks, such as fairness evaluation (as you referenced), sample weighting, and logit adjustment [1][2][3].
>
> Following your helpful suggestion, we have added more details about the rationale behind the selection of prediction characteristics in **Section 3.1.1** of our revised manuscript. The added content is as follows: "*First, we consider the commonly used hallucination detection metrics from previous studies, specifically uncertainty and consistency. Next, we incorporate three additional metrics that reflect prediction confidence and are commonly employed in previous machine learning tasks, such as fairness evaluation and sample weighting (Zhang et al., 2020; Ross & Doll’ar, 2017; Jin et al., 2024).*"
>
> Based on the analyses presented in Section 4.6 of our manuscript, all extracted characteristics contribute effectively to hallucination detection. Their importance, ranked from highest to lowest, is as follows: consistency, uncertainty, margin, probability, and logits norm. Additionally, we do not need to manually assign weights to these characteristics; instead, we learn the function between prediction characteristics and hallucinations automatically using the proposed D3T network.
>
> > **Q4: Model applicability with increasing training complexity, particularly in the context of LoRA.**
>
> To assess the model applicability with increasing training complexity, we analyze how the performance of our approach changes with an increase in the LoRA rank. The results on the NQ dataset, using HADEMIF with fine-tuning, are presented in the table below.
>
> |     Method                       |     acc@50    |     cov@50    |     ECE      |     Brier    |
> |:----------------------------------|:---------------:|:---------------:|:--------------:|:--------------:|
> |     Original   LLM               |     0.288     |     0.115     |     0.171    |     0.196    |
> |     Label   Smoothing            |     0.208     |     0.061     |     0.186    |     0.212    |
> |     Temp.   Scaling              |     0.288     |     0.115     |     0.165    |     0.193    |
> |     LITCAB                       |     0.300     |     0.105     |     0.101    |     0.169    |
> |     LITCAB   w/ Temp. Scaling    |     0.300     |     0.105     |     0.083    |     0.164    |
> |     HADEMIF   (Rank 8)           |     **0.355**     |     0.120     |     **0.026**    |     **0.119**    |
> |     HADEMIF   (Rank 16)          |     0.350     |     0.116     |     0.030    |     0.121    |
> |     HADEMIF   (Rank 32)          |     0.352     |     **0.125**     |     0.029    |     0.120    |
>
> The results show that HADEMIF consistently outperforms other approaches across different rank values, highlighting its applicability. We also observe that using a low rank in LoRA can already achieve strong performance, consistent with the findings presented in the original LoRA paper. These experiments have been added to Appendix A.5 in our revised manuscript, which is highlighted in blue.
>
> Moreover, from the results presented in Tables 6 and 7 of the manuscript, our method steadily demonstrates effectiveness across a wide range of model sizes (from 1.5B to **30B**) and architectures, highlighting its broad effectiveness. To further validate the scalability and versatility of our approach, and to highlight its potential future directions for efficiency improvements, we also include a complexity analysis of the two hallucination detection networks and an evaluation of the computational overhead in **Appendices A.11** and **A.12** of our revised manuscript.
>
> [1] Geometry-aware instance-reweighted adversarial training.
>
> [2] Adaptive logit adjustment loss for long-tailed visual recognition.
>
> [3] Focal loss for dense object detection.

---

> ### Author Response · Authors · 2024-11-23
> **Official Comment by Authors**
>
> Dear Reviewer 76SS:
>
> We hope that our explanations and revisions have adequately addressed your concerns. We sincerely appreciate your thoughtful and insightful feedback, which has been invaluable in enhancing the quality of our paper. Should any points remain unclear, or if you have further questions or suggestions, please do not hesitate to share them. We would be deeply appreciative if you could take our clarifications and improvements into account when reassessing our work.
>
> Thank you once again for your time and effort in reviewing our work.
>
> Sincerely,
>
> Authors

---

> ### Author Response · Authors · 2024-11-26
> **Looking forward to your reply.**
>
> Dear Reviewer 76SS:
>
> We sincerely appreciate the time and effort you have devoted to reviewing our manuscript. As the author-reviewer discussion period comes to a close in the coming week, we kindly inquire if you are satisfied with the clarifications and revisions we have made. Your feedback on the current revision would be extremely valuable, and we would be truly grateful for any further suggestions you may have to provide us with an additional opportunity to refine our work.
>
>
> Once again, we express our heartfelt gratitude for your constructive comments and your contribution to strengthening our submission. Should any concerns remain, we would be more than happy to address them in full.
>
> Best regards,
>
> Authors

---

> ### Author Response · Authors · 2024-12-01
> **Gentle Reminder**
>
> Dear Reviewer 76SS, we sincerely thank you for your thoughtful and thorough review of our manuscript, which has significantly strengthened the quality of our manuscript. As the discussion stage is ending soon, we would like to kindly inquire if you have any remaining questions or concerns. Any feedback is greatly appreciated and will undoubtedly help further enhance our work. Moreover, we would be truly grateful if you would consider increasing the score in light of the revisions and clarifications we have made, which were directly informed by your insightful feedback. Thanks again for your efforts in reviewing our paper!

---

### Official Review · Reviewer_ApSu · 2024-11-01

**Soundness:** 2
**Presentation:** 3
**Contribution:** 3
**Rating:** 6
**Confidence:** 4

**Summary:**

The paper addresses the issue of knowledge hallucinations in large language models (LLMs), which can undermine their reliability in critical domains. The authors propose a framework called HADEMIF to detect and mitigate hallucinations in LLMs using two compact networks: a Deep Dynamic Decision Tree (D3T) and a Multilayer Perceptron (MLP). These networks capture hallucinations in both the output and internal semantic spaces of LLMs, allowing for prediction calibration and improved reliability.

**Strengths:**

Comprehensive Detection: HADEMIF captures hallucinations in both output and internal spaces, providing a more holistic approach compared to previous methods that focus on single aspects.

Interpretable Model: The D3T model offers interpretability, allowing for the identification of key characteristics that impact hallucination detection.

Efficiency: The framework introduces less than 2% additional parameters relative to the LLMs, making it efficient in terms of computational resources.

Versatility: HADEMIF can be applied during both inference and fine-tuning phases, enhancing its applicability across different stages of LLM deployment.

Improved Calibration: The framework significantly reduces the expected calibration error and Brier scores, indicating better alignment between model confidence and accuracy.

**Weaknesses:**

Dependency on Internal States: The approach requires access to the internal states of LLMs, which may not be feasible for black-box models where such access is restricted.

Limited Metric Scope: While the framework considers multiple metrics, the scope is still relatively constrained, potentially limiting detection performance in more complex scenarios.

Complexity in Implementation: The integration of two separate networks (D3T and MLP) may increase the complexity of implementation and maintenance.

**Questions:**

Scalability: How does the framework perform with even larger models or in real-time applications where speed is critical?

Generalizability: Can the framework be adapted to other types of hallucinations beyond those related to factual inaccuracies?

Black-box Applicability: Are there potential adaptations of HADEMIF that could work with black-box models where internal states are not accessible?

Metric Expansion: What additional metrics could be integrated into the framework to enhance its detection capabilities?

User Trust: How does the framework impact user trust in LLMs, and are there ways to quantify this improvement?

---

> ### Author Response · Authors · 2024-11-20
> **Response to Reviewer ApSu (Part I)**
>
> Dear Reviewer ApSu,
>
> We sincerely thank you for your thoughtful review and constructive comments. All the comments have been carefully considered and addressed in our revised manuscript. Below are our detailed responses to your comments:
>
> >**Q1: How does the framework perform with larger models or in real-time applications?**
>
> Our framework is model-agnostic, enabling seamless adaptation to larger models. Moreover, only an additional parameter cost of less than 2% of the total parameters in the LLMs is introduced, ensuring scalability and efficiency.
>
> Experiments have been conducted to evaluate the performance of our approach across models of varying sizes and architectures. As demonstrated in **Tables 6** and **7** of the Appendix, our method consistently delivers strong performance across models ranging from 1.5B to **30B**. In the context of real-time applications, LLMs fine-tuned using our approach can be directly deployed, avoiding an increase in inference time while simultaneously enhancing model calibration and mitigating hallucinations.
>
> >**Q2: Can the framework be adapted to other types of hallucinations?**
>
> The proposed framework incorporates two hallucination detection networks to identify hallucination biases, such as overconfidence and inaccuracies, in both the output and hidden spaces of LLMs. These networks then generate logit adjustment terms that calibrate the predicted probability distribution of the LLMs, aiming to maximize the token probabilities for correct generations while reducing the likelihood of incorrect ones. This approach can mitigate various types of hallucinations, including **factual**, **logical**, and **coherence**-related ones, thereby improving the reliability of the model's generations.
>
> To further evaluate the effectiveness of our approach in hallucination detection, we compared HADEMIF with three additional detection methods, in addition to those presented in our manuscript. The metrics include the area under the receiver operator characteristic curve (AUROC) and Pearson Correlation Coefficient (PCC), both of which are commonly used in previous studies. The comparison results are presented in the table below, where our approach demonstrates superior hallucination detection performance compared to previous studies that rely solely on a single aspect of the metric. These experiments have been included in Appendix A.4 of the revised version of our manuscript (**highlighted in blue**).
>
> |     Dataset                     |     NQ              |                     |                 |     TrivalQA        |                  |              |
> |:---------------------------------|:---------------------:|:---------------------:|:---------------------:|:---------------------:|:---------------------:|:---------------------:|
> |     Metric                      |     AUC$_{s}$       |     AUC$_{r}$       |     PCC         |     AUC$_{s}$       |     AUC$_{r}$    |     PCC      |
> |     Perplexity   [1]            |     0.740           |     0.747           |     0.301       |     0.836           |     0.836        |     0.544    |
> |     LN-Entropy   [2]            |     0.728           |     0.737           |     0.298       |     0.834           |     0.832        |     0.540    |
> |     Lexical   Similarity [3]    |     0.738           |     0.759           |     0.306       |     0.826           |     0.840        |     0.556    |
> |     HADEMIF                     |     **0.772**           |     **0.776**           |     **0.389**       |     **0.843**           |     **0.842**        |     **0.581**    |
>
> >**Q3: Are there potential adaptations of HADEMIF that could work with black-box models?**
>
> Thanks for your good question. We propose two feasible strategies for adapting our approach to black-box models:
>
> - **Knowledge Distillation:** This strategy involves transferring knowledge from a black-box model to a student white-box model. The hidden states of the student model can then be passed into an MLP network for hallucination detection within the semantic space. While this approach may detect the hallucinations in the internal space of both student and teacher models, it offers a possible way to access the internal states of black-box models.
> - **Direct Logits Input:** For a simpler adaptation, one can directly input the logits vector into the MLP network. Although the logits vector does not fully capture the LLMs’ internal states, it serves as an intermediary between the model’s internal and output spaces, reflecting, to some extent, the model’s internal states.
>
> We will explore detailed approaches for applying our method to black-box models in future research.
>
>
> [1] Out-of-distribution detection and selective generation for conditional language models.
>
> [2] Uncertainty estimation in autoregressive structured prediction.
>
> [3] Towards collaborative neural-symbolic graph semantic parsing via uncertainty.

---

> ### Author Response · Authors · 2024-11-20
> **Response to Reviewer ApSu (Part II)**
>
> > **Q4: What additional metrics could be integrated into the framework?**
>
> To strike a balance between efficiency and performance while enhancing the interpretability of D3T, our approach incorporates five types of representative metrics. Comparative and analytical experiments in our manuscript demonstrate their effectiveness. To further improve detection performance, additional metrics may be considered, such as the angles between hidden states and the discriminative weights in the final linear layer for each token, loss gradient, robustness, as well as other statistical properties (e.g., variance) of the logits and probability vectors. Our future research will investigate the incorporation of additional metrics and develop efficient methods for their computation, with the goal of further improving both detection performance and computational efficiency.
>
> We have revised the sentences in the limitations section (Appendix A.10) of our manuscript, which now read as follows: "*Additionally, to further enhance detection performance, future research could explore the integration of additional metrics, such as those related to learning dynamics (e.g., loss gradient) and prediction robustness, and develop efficient computational methods for their evaluation.*"
>
> > **Q5: How does the framework impact user trust in LLMs?**
>
> Our approach enhances user trust in LLMs in two key aspects:
>
> - **Enhancing model calibration and decreasing hallucinations can enhance the users’ trust in the generations of LLMs.** These two aspects can be quantified through a set of metrics that we have adopted. From Tables 1, 2, and 3 in our manuscript, we adopted four metrics, including ECE, Brier, acc@q, and cov@p, to assess the performance of model calibration and hallucination mitigation. The results demonstrate that the performance of our approach consistently surpasses that of the compared baselines. Moreover, we evaluated the hallucination detection performance of our approach using metrics AUROC and PCC. The results presented in Table 8 of our revised manuscript manifest the superior hallucination detection performance of our approach.
> - **The enhancement of interpretability can also improve user trust.** Using D3T as the hallucination detection network helps to reveal the hallucination detection rules (Appendix A.7), showing how different combinations of characteristics determine the existence of hallucinations, and it can also uncover the importance of various prediction characteristics in the hallucination detection process (Section 4.6). This enhances decision transparency, providing users with a deeper understanding of the existence of hallucination in LLMs, thereby further increasing their trust in the model's decisions when applying it.
>
> > **Q6: The integration of two separate networks may increase complexity.**
>
> We recognize that the computational complexity is increased, which is, in fact, an inherent challenge when incorporating additional modules, such as adapters in parameter-efficient fine-tuning of LLMs, as well as in the LITCAB method. Nevertheless, we have analyzed the complexity of the two hallucination detection networks in Appendix A.11, demonstrating that these networks are parameter efficient and exhibit strong scalability. Furthermore, we expanded the discussion on computational costs in Appendix A.12 of the revised manuscript. Our analysis reveals that using Llama2-7B as a case study, the time required to update the two hallucination detection networks per iteration is only **0.9%** of the time needed to update the LLMs with LoRA (rank = 8).

---

> ### Author Response · Authors · 2024-11-23
> **Official Comment by Authors**
>
> Dear Reviewer ApSu:
>
> We hope that the revisions and clarifications have effectively addressed your concerns. Your insightful feedback has greatly contributed to enhancing the quality of our paper. If any aspects remain unclear or if you have additional questions or suggestions, please feel free to let us know. We deeply value your comments and would be truly grateful if you could consider incorporating these improvements when reassessing our work.
>
> Thank you once again for your time and effort in reviewing our work.
>
> Sincerely,
>
> Authors

---

> ### Author Response · Authors · 2024-11-26
> **Looking forward to your reply.**
>
> Dear Reviewer ApSu:
>
> We sincerely appreciate the time and effort you have dedicated to reviewing our manuscript. As the author-reviewer discussion period comes to a close in the coming week, we kindly inquire if you are satisfied with the clarifications and revisions we have made. Your feedback on the current revision would be extremely valuable, and we would be truly grateful for any further suggestions you may have to provide us with an additional opportunity to refine our work.
>
>
> Once again, we would like to express our heartfelt gratitude for your thoughtful and constructive review, which has been instrumental in strengthening our submission. Should any further concerns remain, we are fully committed to addressing them.
>
> Best regards,
>
> Authors

---

> ### Author Response · Authors · 2024-12-01
> **Gentle Reminder**
>
> Dear Reviewer ApSu, we sincerely thank you for your thoughtful and thorough review of our manuscript, which has significantly strengthened the quality of our manuscript. As the discussion stage is ending soon, we would like to kindly inquire if you have any remaining questions or concerns. Any feedback is greatly appreciated and will undoubtedly help further enhance our work. Moreover, we would be truly grateful if you would consider increasing the score in light of the revisions and clarifications we have made, which were directly informed by your insightful feedback. Thanks again for your efforts in reviewing our paper!

---

### Author Response · Authors · 2024-12-03
**Official Comment by Authors**

We sincerely appreciate the time and efforts of all the reviewers throughout the review process and their insightful and valuable feedback, which are instrumental in helping us refine and enhance our manuscript. We are deeply grateful and pleased that all reviewers have recommended the acceptance of our work. They recognized that our study is novel and well-motivated (Reviewers 76SS and x16Z), that the proposed method is comprehensive, interpretable, and parameter-efficient with broad applicability (Reviewers ApSu, QEQq, and x16Z), that the experimental investigation is extensive with promising performance (Reviewers ApSu, 76SS, QEQq, x16Z, and goPV), and that the paper is easy to follow (Reviewers goPV and 76SS). We have carefully considered and addressed each of the reviewers' concerns by providing further clarifications and conducting the requested experiments in our revised manuscript. Once again, we would like to express our sincere gratitude to all the reviewers for their invaluable contributions.

---

### Meta-Review · Area_Chair_t5U8 · 2024-12-24

**Metareview:**

This paper tackles the problem of LLM hallucination: its identification/detection and mitigation/reduction. The proposed framework, HaDeMiF, applies two compact networks, an interpretable Deep Dynamic Decision Tree (D3T) in the output space, and a MLP in the internal hidden states. The framework is applicable in both inference and fine-tuning stages, and experiments demonstrate its superior performance in hallucination detection and calibration across various open-source LLMs compared to existing methods.

The reviewers all rated towards accepting the work, recognizing that the solution is novel and parameter-efficient with broad applicability, and the study comprehensive. A majority were convinced to through a rather successful rebuttal.

**Additional Comments On Reviewer Discussion:**

The author did a great job at rebuttal. They were highly responsive, respectful, and quick at addressing reviewers' concerns with new experimental results. 3 out of 5 reviewers responded with a raised score. The other 2 never engaged, but it's expected that they'd find the rebuttal convincing.

---

### Decision · Program_Chairs · 2025-01-22

Accept (Poster)